# SCREBOUND: AN EFFICIENT DESIGN OF SINGLE-CELL FOUNDATION MODEL WITH BATCH REPRESENTATION

## ABSTRACT

Recent advances in single-cell foundation models (scFMs) have demonstrated the promise of large-scale pretraining on single-cell RNA sequencing (scRNA-seq) data for a wide range of downstream biological tasks. However, existing models such as scGPT, UCE, scFoundation, and scMulan demand substantial computational resources for both training and inference, limiting their accessibility and practical deployment in academic settings. Furthermore, the systematic noise within different experimental batches of scRNA-seq datasets, also termed as batch effect, cannot be well removed with the masked token prediction tasks that are commonly used by these models. This significantly jeopardizes the zero-shot performance of these models on new data experiments. In this work, we present a novel and efficient design for single-cell foundation models that significantly reduces computational costs while improving the robustness of cell representation learning. Our architecture introduces a biologically-informed compression strategy to reduce input token numbers of each cell without sacrificing key transcriptomic signals. We also proposed a novel biologically-informed batch encoding strategy and introduced a multi-granular supervised contrastive loss to account for the batch effect during the model pre-training phase. We validate our design through extensive experiments across diverse datasets, demonstrating competitive performance in key zero-shot tasks including cell type annotation, batch effect removal, cross-species knowledge transfer, and missing value imputation, while achieving up to 17x reduction in inference time and 30x reduction of memory usage compared to the SOTA model scGPT. Our design makes foundational single-cell modeling more accessible and robust.

## 1 INTRODUCTION

Single-cell RNA-sequencing (scRNA-seq) technology measures the gene expression levels of genome-wide genes in each single cell, and a scRNA-seq experiment can measure thousands or even millions of cells simultaneously. Recently, large-scale scRNA-seq atlases, such as the Human Cell Atlas (Regev et al., 2017), the Human Ensembled Cell Atlas (Chen et al., 2022), and CELLxGENE (Program et al., 2023), have been constructed by collecting data from many experiments. These atlases provide single-cell profiles across diverse tissues, species, and experimental conditions, enabling the training of foundation models tailored to single-cell biology.

Single-cell foundation models (scFMs) are trained on these large-scale datasets to generate unified representations of cells for various downstream biological analyses. Existing scFMs such as scGPT (Cui et al., 2024), UCE (Rosen et al., 2023), scFoundation (Hao et al., 2024), Geneformer (Theodoris et al., 2023) and scMulan (Bian et al., 2024a) have demonstrated promising performance in various downstream tasks. However, a critical bottleneck remains: the substantial computational demands of training and deploying these models limit their accessibility and adoption, particularly in academic settings.

In addition, scRNA-seq data obtained from different experiments are generated with different experimental environments, handling personal, and sequencing protocols (Luecken & Theis, 2019; Zhang et al., 2024), leading to severe distributional shifts across experimental batches – even when measuring similar tissue samples. These systematic distortions, known as batch effects, makes it

difficult to integrate scRNA-seq data from different experimental sources. Traditional scFM pre-training tasks, such as masked expression prediction, are not designed to handle these distortions, as the prediction target, which is the masked gene expression, contains both biological signals and unwanted batch-related noise. Existing scFMs either do not explicitly model batch effects (Rosen et al., 2023; Bian et al., 2024b), or use isolated one-hot vectors to represent batch IDs (Richter et al., 2025), which lack contextual information about the batches and cause problems when the test data is from an unseen batch.

In this work, we tackle both challenges by proposing a new method, named scREBOUND (Single Cell **RE**presentation of **B**atch and **O**mics **U**sing efficie**N**t **D**esign). First, we introduce a biologically-informed gene compression network that reduces the total number of gene tokens per cell. The compression network groups genes with similar functionality, informed by their protein embeddings from a protein language model, and is fine-tuned during training to better retain the expression information in the original data. This compression significantly accelerates training and inference, while reducing GPU memory usage. Second, we address batch effects by incorporating a batch embedding network and a multi-granular contrastive loss as regularization during model pretraining. The batch embedding network learns representations from batch-specific features summarized from expression data of certain categories of genes and can generalize to unseen batches. The multi-granular contrastive loss align cells of common cell type across batches and is adjusted to accommodate the label granularity mismatch across different experiments.

We evaluate our model on multiple single-cell datasets, under four zero-shot tasks: (1) Batch effect removal: given a test dataset, learns representations of cells such that batch effects are excluded and cells of the same type from different batches align in the latent space; (2) Cell type annotation: given a test scRNA-seq dataset, predicts the cell type label of each cell in the test data; (3) Cross-species knowledge transfer: given a test data from species unseen in the training dataset, learns cell representation that still preserves cell-cell variation; (4) Gene expression imputation: given a test dataset with only partial gene expression profiles, predicts the expression levels of the missing genes. We compared scREBOUND with state-of-the-art (SOTA) scFMs. Our results demonstrate that scREBOUND achieves comparable or superior zero-shot performance across all four tasks, while significantly reducing training and inference costs. This work paves the way for enabling wider adoption of scFMs in academic research and improving the robustness of single-cell analyses across heterogeneous datasets.

We summarize our contributions of this work as follows:

- We proposed scREBOUND, a single-cell foundation model that is computationally efficient and better accommodates the batch effect existing across single-cell experiments.

- We introduced a protein-language-model-informed gene compression strategy that reduces the number of gene tokens per cell while preserving key biological information. We provided theoretical analysis on the efficiency of the compression network from information theory perspective.

- We also introduced a batch embedding network and multi-granular contrastive loss regularization to account for the batch effect during model pre-training.

- We conducted comprehensive evaluations of scREBOUND on four zero-shot single-cell tasks and showed that it matches or surpasses the performance of existing scFMs while substantially lowering computational requirements.

## 2 RELATED WORK

Recently, several scFMs have been proposed to leverage large-scale scRNA-seq atlases for cell representation learning, which is then used for various downstream tasks, including cell type annotation, batch effect removal and gene expression imputation (Cui et al., 2024; Hao et al., 2024; Bian et al., 2024b; Rosen et al., 2023; Theodoris et al., 2023). Most of these models are constructed using transformer-based architecture, treating each cell as a "cell sentence" and a gene as "word token" encoding gene identity and its expression level. However, unlike natural language or time-series data, genes do not have a natural order within each cell, making the direct application of transformer models to scRNA-seq data nontrivial. Due to this lack of inherent ordering, most existing scFMs adopt masked token prediction as the pretraining objective (Bian et al., 2024b). Details on token design and training objectives of existing scFMs are in Appendix Sec. A.1.

Despite existing developments, scFMs remain an evolving area with critical design challenges. Human genome includes around 20k protein-coding genes, and most of the gene expression are 0s in scRNA-seq data due to low molecule capture rate. This make scRNA-seq data extremely sparse and noisy. To reduce noise and improve efficiency, most models (Bian et al., 2024b; Cui et al., 2024) select the top 10–20% highly variable genes (HVGs) for training. However, HVG selection can introduce data-dependent biases (Zhao et al., 2024) and substantial information loss. Moreover, even top 10–20% HVGs typically include thousands of genes, causing computational burden of the scFMs. Moreover, recent benchmarking efforts (Liu et al., 2023; Kedzierska et al., 2023) questioned the ability of scFMs in batch effect removal ability and other downstream tasks, highlighting the need for models that are both computationally efficient and more effective in handling critical biological variability.

## 3 Efficient Foundation Modeling with Improved Representation Learning

### 3.1 Overview

scREBOUND uses an encoder-only transformer architecture with masked token prediction (MTP) pretraining task to learn the cell representation from scRNA-seq data. We consider that the large number of gene tokens is one of the main factors that cause computational cost in existing scFMs. In scREBOUND, we take an approach to reduce the number of tokens that is both biology-informed and efficient. We introduce a compression module (Fig. 1a) to map all input genes to 256 meta-genes. This process is explained in detail in Sec. 3.2. The compressed meta-gene token embedding and expression embedding are then fed into the transformer module (Fig. 1b), which reduces the number of tokens processed by the transformer by $\sim$5 times compared to UCE, and by $\sim$70 time compared to scFoundation.

In the transformer module (Fig. 1b), we incorporate two designs to mitigate batch effects in the training data. First, we use a batch encoder, which takes batch-specific features and learns a batch representation, serving as the batch condition to adjust for the prediction result of mask predictor network. Then, we take advantage of the widely-available cell type labels of cells in the training data, as additional supervision to align cells of the same cell types but from different batches. However, a challenge is that the cell type labels obtained from different batches (usually from different sources) may be at different granularities, where cells of a certain cell type (e.g. T cell) may have a detailed sub-type annotation in one experiment (e.g. CD4+ T cell) and a coarse annotation in another experiment (e.g. immune cell). We designed a multi-granular contrastive loss, that accommodates the multi-granular nature of cell type labels, as a regularizer during training to correct for batch effects.

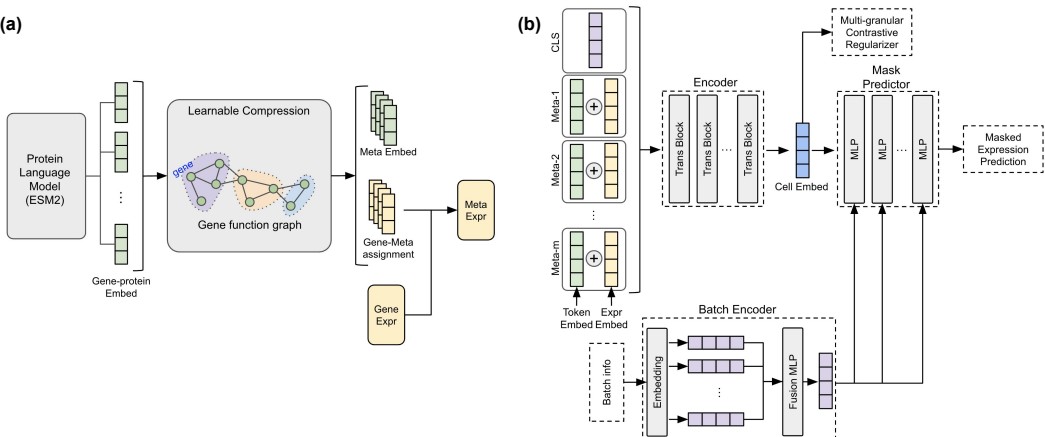

Figure 1: The graphical pipeline of scREBOUND. **a**. The gene compression module. **b**. The foundation model network (transformer module). Detailed architecture is in Appendix Sec. A.2.

### 3.2 Functional gene compression guided by protein language model

A scRNA-seq experiment generates a continuous "cell by gene" expression matrix (denoted by $\mathbf{E}$), and element $e_{i,j}$ in the matrix records the expression level of gene $j$ in cell $i$. We denote the number of genes in $\mathbf{E}$ by $n$.

For each gene, we used the corresponding protein embedding obtained from a protein language model (ESM2 (Lin et al., 2022)) as functional representation of the gene's identity. This allows us to further perform biologically-meaningful compression of gene features, as compression on such gene-protein embedding allows for grouping of functionally similar genes and induces less biological information loss. To learn the gene compression, denoting the protein embedding of gene $j$ learned by ESM2 by $\mathbf{p}_j$, we construct a k-nearest neighbor graph $\mathbf{G} \in \mathbb{R}^{n \times n} (k = 5)$ using the pairwise distance between protein embeddings $\{\mathbf{p}_j\}_{j=1}^n$. Given $n$ genes and $m$ meta-genes ($m = 256$), we define a learnable "gene to meta-gene" soft assignment matrix $\mathbf{S} \in \mathbb{R}^{n \times m}$. $\mathbf{S}$ is row-wise softmax-ed to encode the gene to meta-gene assignment probability: $\mathbf{S}_{j,:} = \text{softmax}(\mathbf{S}_{j,:}/\tau_s)$ ($\tau_s I = 0.2$). The gene compression module learns $\mathbf{S}$ through the training process and regularize $\mathbf{S}$ with protein function knowledge graph $\mathbf{G}$ through a mincut regularization loss (Bianchi et al., 2020)

$$\mathcal{L}_G(\mathbf{S}, \mathbf{G}) = \frac{Tr(\mathbf{S}^T \mathbf{A} \mathbf{S})}{Tr(\mathbf{S}^T \mathbf{G} \mathbf{S})} + \lambda_{orth} \|\widetilde{\mathbf{S}}^T \widetilde{\mathbf{S}} - \mathbf{I}\|_2^2 \tag{1}$$

where $\mathbf{A}$ is the degree matrix of $\mathbf{G}$, and $\widetilde{\mathbf{S}}$ is the column-normalized $\mathbf{S}$. The first term $\frac{Tr(\mathbf{S}^T \mathbf{A} \mathbf{S})}{Tr(\mathbf{S}^T \mathbf{G} \mathbf{S})}$ is a relaxed-mincut loss that forces $\mathbf{S}$ to preserve structural information from graph $\mathbf{G}$, whereas $\|\widetilde{\mathbf{S}}^T \widetilde{\mathbf{S}} - \mathbf{I}\|_2^2$ regularizes the mutual exclusiveness between meta-genes ($\lambda_{ortho} = 0.01$).

For each cell $i$, the compression model generates meta-gene expression $e_{i,m_j}$ and token embedding $\mathbf{p}_{m_j}$ using $\mathbf{S}$ (Fig. 1a):

$$e_{i,m_j} = \sum_{j=1}^n e_{i,j} \mathbf{S}_{j,m_j}, \quad \mathbf{p}_{m_j} = \sum_{j=1}^n \mathbf{p}_j \mathbf{S}_{j,m_j} \tag{2}$$

The efficiency of the mincut regularized compression strategy in scREBOUND is theoretically analyzed from information theory perspective (Appendix Sec. A.3).

### 3.3 Tokenization and pretraining objective of scREBOUND

After obtaining the meta-genes' expression $e_{i,m_j}$ and token embedding $\mathbf{p}_{m_j}$, we construct meta-gene sentence $c_i$ for cell $i$ with sequence:

$$c_i = [\text{cls}, [e_{i,m_1}, \mathbf{p}_{m_1}], [e_{i,m_2}, \mathbf{p}_{m_2}], \cdots, [e_{i,m}, \mathbf{p}_m]] \tag{3}$$

the "cls" token embedding is randomly initialized, and used as the position for the cell embedding generation of scREBOUND. There are three major ways to transform a continuous value $e_{i,m_j}$ into embedding: (1) Digitizing $e_{i,m_j}$ and learns embedding for each digit; (2) Feeding $e_{i,m_j}$ into Multilayer Perceptron (MLP); and (3) Feeding $e_{i,m_j}$ into Fourier encoding function (Vaswani et al., 2017). Method (1) prevents the gradient from back-propagating to the gene compression model, and method (2) does not work well with the long-tail gene expression level distribution (Lopez et al., 2018). Therefore, scREBOUND adopts Fourier encoding function to transform $e_{i,j}$ into embedding. scREBOUND then preprocessed the token and value embeddings using MLP and summed up the two embeddings ("Token Embed" and "Expr Embed" in Fig. 1b) element-wise, to obtain the final token embedding for the construction of $c_i$ (Equ. 3). The tokenized $c_i$ is then fed into the transformers (Fig. 1b) to generate the embedding $\mathbf{h}_i$ of cell $i$.

scREBOUND is pretrained on the masked token prediction (MTP) task. For each training cell, we randomly drop the expression of genes (by setting the expression value "0") with probability dynamically changing from $0.1$ to $0.4$, and feed the "zero-masked" expression into scREBOUND which then predict the dropped gene expression value. Given the long-tailed nature of data distribution, we used negative binominal likelihood loss (NB loss) to measure model prediction accuracy (Lopez et al., 2018). Given the ground truth expression $e_{i,j}$ of gene $j$ in cell $i$, the model estimated mean $\hat{e}_{i,j}$ and dispersion $\theta_{i,j}$ is modeled with distribution (Lopez et al., 2018):

$$NB(e_{i,j}; \hat{e}_{i,j}, \hat{\theta}_{i,j}) = \frac{\Gamma(e_{i,j} + \hat{\theta}_{i,j})}{\Gamma(\hat{\theta}_{i,j})} \left( \frac{\hat{\theta}_{i,j}}{\hat{\theta}_{i,j} + \hat{e}_{i,j}} \right)^{\hat{\theta}_{i,j}} \left( \frac{\hat{e}_{i,j}}{\hat{\theta}_{i,j} + \hat{e}_{i,j}} \right)^{e_{i,j}} \tag{4}$$

scREBOUND is then trained to estimated the $\hat{e}_{i,j}, \hat{\theta}_{i,j}$ for the masked gene $j$s by minimizing the negative log-likelihood of Equ. 4.

### 3.4 Contrastive regularization with multi-granular labels

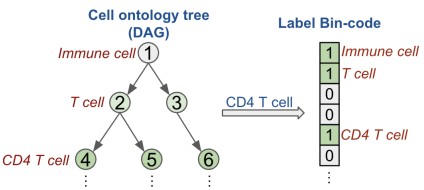

Figure 2: Transform cell label into hierarchical binary code

To remove batch effect, we constructed multi-granular contrastive loss to align cells of same cell type across batches. To account for the mismatch in granularity of cell type annotations (detailed in Sec. 3.1) provided by different data sources, we transform the annotation into binary code (bin-code) based on the cell ontology tree (Diehl et al., 2016), which provides cell label hierarchy information (Fig. 2). Details on bin-code construction are in Appendix Sec. A.4. The cell type bin-codes make annotations comparable across data batches, e.g., cell type labels that have bin-codes with more common "1"s are considered more closely related.

We then constructed a multi-granular contrastive loss from the label bin-codes of the cells. Given each cell $i$ (termed anchor cell), we put the remaining cells (target cells) into three pools. (1) Positive pool $\mathbb{P}(i)$: target cells of with labels of same type or higher label granularity (same & descendent labels). (2) Neutral pool $\mathbb{M}(i)$: target cells with lower granularity labels that are not guaranteed to be the same type (ancestor labels). (3) Negative pool $\mathbb{N}(i)$: the remaining target cells, these cells have labels of different types (See Appendix Sec. A.4 for the pool calculation from bin-codes).

For every cell $i$, we construct the contrastive loss to act on the embedding space, attracting cells in $\mathbb{P}(i)$, repelling cells in $\mathbb{N}(i)$, and leaving cells in $\mathbb{M}(i)$ not penalized. Given the cell embedding $\{\mathbf{h}_i, \forall i\}$, the loss follows (Khosla et al., 2020):

$$\mathcal{L}_{\mathrm{contr}}(\mathbf{h}_i) = -\log \frac{1}{|\mathbb{P}(i)|} \sum_{j \in P(i)} \frac{\exp(\widetilde{\mathbf{h}}_i^T \widetilde{\mathbf{h}}_j / \tau)}{\sum_{j \in N(i)} \exp(\widetilde{\mathbf{h}}_i^T \widetilde{\mathbf{h}}_j / \tau)} \tag{5}$$

$|\mathbb{P}(i)|$ denotes the size of positive pool for cell $i$, and $\tau$ is the temperature parameter ($\tau = 0.07$).

To prevent potential interference between the masked expression prediction task and the contrastive regularization objective, we introduce a batch encoder module. This module takes as input a set of batch-specific features from the scRNA-seq data, including the proportion of mitochondrial gene counts, the fraction of nonzero-expressing genes, and average expression levels of housekeeping genes, etc (full feature list is in Appendix Table 7). The batch encoder learns the batch representation which is then inserted to the masked predictor module for conditioned prediction (Fig. 1b). The batch encoder module is trained jointly with the remaining part of the foundation model throughout the training process.

### 3.5 PRETRAINING PROCESS

To ensure the stability of the training process, we use a three-step training strategy.

In **Step 1**, we freeze the parameters of the gene compression module (**S** properly initialized by minimizing Equ. 1, details see Appendix Sec. A.6), and train the remaining networks of scREBOUND jointly through MTP task. Since the gene compression module is fixed, we directly apply random "zero-mask" on the meta-gene expression and train scREBOUND to predict the masked meta-gene expression by minimizing the MSE loss. For each cell $i$, MSE loss $\mathcal{L}_{s1}(i)$ is calculated between the ground truth meta-gene expression $e_{i,m_j}$ and predicted expression $\hat{e}_{i,m_j}$ for gene $j$ in mask meta-gene set $M_i$:

$$\mathcal{L}_{s1}(i) = \sum_{m_j \in M_i} MSE(\hat{e}_{i,m_j}; e_{i,m_j}) \tag{6}$$

In **Step 2**, we unfreeze gene compression module, and train scREBOUND jointly through MTP task with the random "zero-masking" applied to the original gene expression data. For each cell $i$, the loss is calculated using the NB loss between prediction $\{\hat{e}_{i,j}, \hat{\theta}_{i,j}\}$ and ground truth $e_{i,j}$ for gene $j$ in mask gene set $M_i$ (Sec. 3.3), and regularized by $\mathcal{L}_G$ (Sec. 3.2):

$$\mathcal{L}_{s2}(i) = -\sum_{j \in M_i} \log NB(e_{i,j}; \hat{e}_{i,j}, \hat{\theta}_{i,j}) + \mathcal{L}_G(\mathbf{S}, \mathbf{G}) \tag{7}$$

In **Step 3**, we incorporate the multi-granular contrastive regularization (Sec. 3.4), and train the model with both $\mathcal{L}_{s2}(i)$ and the contrastive regularization:

$$\mathcal{L}_3(i) = \mathcal{L}_{s2}(i) + \mathcal{L}_{\text{contr}}(\mathbf{h}_i) \qquad (8)$$

## 4 EXPERIMENTAL RESULTS

We trained scREBOUND using 35 million healthy human single cells obtained from the CELLxGENE human cell collection (Program et al., 2023)(Ver. 2024-07-01). We used 8 human scRNA-seq datasets that covers various application scenarios for model evaluations. The test datasets include (1) 3 standard multi-batch datasets used in scIB benchmarking platform (Luecken et al., 2022): human immune cell dataset (named "Immune"), human pancreas cell dataset (named "Pancreas"), human lung atlas (named "Lung"); (2) 1 large-scale covid-19 dataset (named "Covid19") gathered from 3 independent studies (Arunachalam et al., 2020; Lee et al., 2020; Wilk et al., 2020; Zhang et al., 2024); (3) 4 standard multi-batch datasets used in scEval benchmarking platform (Li et al., 2025) (named "Cell Line", "PBMC 1", "Pancrm", "PBMC 2"). The sizes of these datasets are shown in Table 1.

Table 1: Overview of datasets.

| Dataset | #cells | #batches |
|---|---|---|
| CELLxGENE | 35M | 10461 |
| Immune | 33,506 | 10 |
| Pancreas | 16,382 | 9 |
| Lung | 32,472 | 16 |
| Covid19 | 163,729 | 41 |
| Cell Line | 8891 | 3 |
| PBMC 1 | 15476 | 2 |
| Pancrm | 11125 | 5 |
| PBMC 2 | 4638 | 2 |

In the following, we first show scREBOUND's efficiency in running time and memory usage compared to baseline methods, then we show that scREBOUND achieves better or comparable zero-shot performance on 4 downstream tasks covering various single-cell application scenarios: (1) batch effect removal, (2) cell type annotation, (3) cross-species knowledge transfer, and (4) missing expression imputation. We compare scREBOUND with baseline scFMs: scGPT (Cui et al., 2024), scMulan (Bian et al., 2024a), UCE (Rosen et al., 2023), and scFoundation (Hao et al., 2024). scGPT has two models, and we use the scGPT-continual-pretrained (scGPT-continual) model as it has better performance in zero-shot cell embedding related tasks. UCE has two models, the 4-layer and 33-layer models, and we use the 33-layer model which is expected to have better performance for the comparison. For other baseline methods, we use the default modes. For the last downstream task, gene expression imputation, we used only scGPT and scFoundation out of all scFMs as other scFMs cannot perform this task. To include more baseline methods for this task, we used two models of scGPT, and added a non-scFM method scVI (Lopez et al., 2018), which is a widely used task-specific variational auto-encoder (VAE) method. We do not include Geneformer (Theodoris et al., 2023) in our benchmarking (except for running time and memory usage comparison) because it requires fine-tuning and tasks we evaluate are zero-shot.

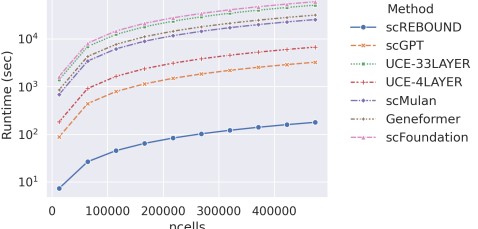

Figure 3: Running time comparison of scRE-BOUND and baseline methods.

Table 2: Running time and memory usage comparison between scREBOUND and baseline methods.

| Methods | ncells | runtime (s) | peak memory (GB) |
|---|---|---|---|
| scREBOUND | 128 | **0.05** | **0.56** |
| scGPT | 128 | 0.88 | 17.70 |
| UCE-33LAYER | 128 | 13.89 | 15.91 |
| UCE-4LAYER | 128 | 13.73 | 1.82 |
| scMulan | 128 | 6.91 | 1.54 |
| Geneformer | 128 | 8.57 | 32.31 |
| scFoundation | 128 | 16.37 | 25.07 |

### 4.1 INFERENCE RUNNING TIME AND MEMORY USAGE COMPARISON

We measure the running time and peak memory usage of all methods using varying sizes of inference data. We sampled 500,000 cells from our training dataset (CELLxGENE data) and infer the cell embedding of 128 cells per mini-batch. We ran all methods using one A40 GPU with 48 GB of GDDR6 memory. The cumulative running time comparison on varying sizes of inferred data is shown in Fig. 3. Table 2 shows the average running time of 128 cells for different methods, along with the peak memory usage of different methods during inference. scREBOUND has significant runtime and GPU memory consumption advantage compared to other baseline methods. scGPT is the

second fast method after scREBOUND, and scMulan has the second smallest memory consumption. scFoundation is the slowest method and Geneformer has the largest memory consumption.

## 4.2 TEST ON ZERO-SHOT BATCH EFFECT REMOVAL FOR MULTI-BATCH SCRNA-SEQ DATA

**Experiment setup and evaluation metrics**   We evaluate the zero-shot batch effect removal performance of scREBOUND and baseline methods using aforementioned evaluation datasets. We obtained the cell embedding from the pre-trained models in a zero-shot manner and measured the alignment of the same cell type across batches within the embedding space.

We use the following metrics: ARI (Adjusted Rand Index), NMI (Normalized Mutual Information) and ASW (Average Silhouette Width). In an ideal scenario where batch effects are completely removed, a cell clustering algorithm should only cluster cell representations by cell type identity instead of the batch identity. We follow the process used in a previous benchmarking study on batch integration methods, scIB (Luecken et al., 2022), to calculate these metrics. First, we cluster the cells in the representation space using the Leiden clustering algorithm (Traag et al., 2019a), and measure the agreement between the resulting cluster labels and known cell type labels through ARI and NMI scores. For cells of each ground truth cell type, ASW measures how well cell embedding are aligned across batches. Further details on performing clustering and formulas for calculating ARI, NMI and ASW are in Appendix Sec. A.7.1. For all three metrics, higher is better.

Table 3: Performance of scFMs on batch effect removal. Numbers in bold denote the best, and with underlines denote the second best.

| Methods | scREBOUND | | | scGPT | | | scMulan | | | UCE | | | scFoundation | | |
|---|---|---|---|---|---|---|---|---|---|---|---|---|---|---|---|
| | ARI | NMI | ASW | ARI | NMI | ASW | ARI | NMI | ASW | ARI | NMI | ASW | ARI | NMI | ASW |
| Immune | **0.57** | **0.75** | **0.61** | 0.48 | 0.68 | 0.56 | 0.34 | 0.59 | 0.55 | 0.37 | 0.65 | 0.55 | 0.34 | 0.66 | 0.57 |
| Pancreas | **0.60** | **0.74** | **0.60** | 0.37 | 0.53 | 0.52 | 0.20 | 0.43 | 0.48 | 0.45 | 0.62 | 0.55 | 0.37 | 0.57 | 0.53 |
| Lung | **0.52** | 0.70 | **0.58** | 0.49 | **0.73** | 0.57 | 0.37 | 0.66 | 0.55 | 0.45 | 0.67 | 0.55 | 0.39 | 0.72 | 0.55 |
| Covid19 | 0.46 | **0.63** | **0.64** | **0.61** | **0.63** | 0.56 | 0.26 | 0.53 | 0.58 | 0.29 | 0.57 | 0.57 | 0.24 | 0.55 | 0.56 |
| Cell Line | **0.96** | **0.91** | **0.80** | 0.91 | 0.84 | 0.58 | 0.30 | 0.46 | 0.65 | 0.53 | 0.61 | 0.63 | 0.71 | 0.74 | 0.79 |
| PBMC 1 | **0.57** | **0.69** | **0.65** | 0.53 | 0.63 | 0.51 | 0.27 | 0.57 | 0.58 | 0.44 | **0.69** | 0.58 | 0.39 | 0.67 | 0.58 |
| Pancrm | **0.66** | **0.75** | **0.60** | 0.34 | 0.44 | 0.49 | 0.29 | 0.60 | 0.51 | 0.39 | 0.64 | 0.19 | 0.29 | 0.60 | 0.54 |
| PBMC 2 | **0.81** | **0.79** | **0.68** | 0.41 | 0.53 | 0.51 | 0.37 | 0.63 | 0.59 | 0.42 | 0.57 | 0.52 | 0.55 | 0.72 | 0.59 |

**Results**   The scores are shown in Table 3. Some visualizations of the cell representations using UMAP (McInnes et al., 2018) are shown in Appendix Sec. A.10. scREBOUND shows top performance in all 8 test datasets, thanks to the contrastive regularizer and the accompanying batch encoder, while scGPT has the second-best performance. scFoundation and scMulan have relatively unstable performance especially on Pancreas and Covid19 datasets.

## 4.3 TEST ON ZERO-SHOT CELL TYPE ANNOTATION

**Experiment setup and evaluation metrics**   To perform zero-shot cell type annotation of a given test dataset, we first need to train a classifier that can output cell type labels for cells in the test dataset. We use two simple choices for the classifier, one is a k-nearest neighbor (kNN) classifier and the other is a support vector machine (SVM) classifier. We compare the performance of the methods using both classifiers. The classifiers are constructed using python package "scikit-learn" (Pedregosa et al., 2011). We construct kNN classifier with 5 nearest neighbors, and SVM with "rbf" kernel and class_weight = "balanced".

To train the classifier, we randomly selected 1 million cells from the pretraining dataset (the CELLx-GENE data), and generated the cell embedding using scREBOUND and baseline models. Since the embedding dimensions of some models are too high for training the classifier (scFoundation has 3074 dimensions, and UCE has 1280 dimensions), we reduced the cell embedding of all models to 100 dimensions using PCA. We then trained classifiers separately using the cell embedding obtained from different models, and evaluated the classification accuracy on 4 aforementioned evaluation datasets.

To quantify the performance of cell type annotations, we use two metrics, the weighted F1 score (WF1) and accuracy (Accu). Weighted F1 score is the cluster-size-weighted average F1 score (details

see Appendix Sec. A.7.2), and accuracy is the proportion of correctly predicted cell labels out of all labels.

**Results** The scores are shown in Table 4. scREBOUND shows top performance in the majority of the datasets using both classifiers. scMulan shows good performance in Lung and Covid19 test, but its performance is very low in the Pancreas dataset. The overall superior performance of scREBOUND in cell type annotation also owes to its improved batch effect removal capability over baseline methods.

Table 4: Performance of scFMs on cell type annotation using kNN and SVM classifiers. Numbers in bold denote the best, and with underlines denote the second best.

| Classifier | Methods | Immune | | Pancreas | | Lung | | Covid19 | | CellLine | | PBMC1 | | Pancrm | | PBMC2 | |
|---|---|---|---|---|---|---|---|---|---|---|---|---|---|---|---|---|---|
| | | WF1 | Accu | WF1 | Accu | WF1 | Accu | WF1 | Accu | WF1 | Accu | WF1 | Accu | WF1 | Accu | WF1 | Accu |
| kNN | scREBOUND | **0.74** | **0.72** | **0.80** | **0.81** | 0.52 | 0.59 | 0.71 | **0.73** | **0.99** | **0.99** | 0.85 | 0.85 | 0.89 | **0.89** | **0.92** | **0.92** |
| | scGPT | 0.60 | 0.60 | 0.65 | 0.68 | 0.56 | **0.63** | **0.72** | **0.73** | 0.96 | 0.95 | 0.72 | 0.74 | 0.53 | 0.52 | 0.71 | 0.74 |
| | scMulan | 0.66 | 0.67 | 0.06 | 0.10 | 0.56 | **0.63** | 0.68 | 0.70 | **0.99** | **0.99** | 0.81 | 0.82 | 0.75 | 0.77 | **0.92** | **0.92** |
| | UCE | 0.47 | 0.49 | 0.78 | 0.80 | **0.57** | 0.61 | 0.68 | 0.69 | 0.98 | 0.98 | **0.96** | **0.96** | **0.90** | 0.88 | 0.74 | 0.75 |
| | scFoundation | 0.70 | 0.70 | 0.71 | 0.74 | 0.55 | 0.63 | 0.69 | 0.70 | **0.99** | **0.99** | **0.96** | **0.96** | 0.79 | 0.78 | **0.92** | **0.92** |
| SVM | scREBOUND | **0.78** | **0.79** | **0.90** | **0.89** | 0.54 | 0.58 | 0.79 | 0.79 | **0.99** | **0.99** | 0.84 | 0.83 | **0.92** | **0.90** | 0.91 | 0.91 |
| | scGPT | 0.72 | 0.71 | 0.76 | 0.77 | 0.56 | 0.60 | 0.80 | 0.81 | 0.98 | 0.98 | 0.80 | 0.80 | 0.62 | 0.61 | 0.80 | 0.80 |
| | scMulan | 0.70 | 0.71 | 0.19 | 0.28 | **0.59** | **0.63** | **0.84** | **0.84** | **0.99** | **0.99** | 0.92 | 0.92 | 0.86 | 0.86 | **0.93** | **0.93** |
| | UCE | 0.59 | 0.58 | 0.86 | 0.85 | 0.58 | **0.63** | 0.76 | 0.77 | 0.98 | 0.97 | **0.97** | **0.97** | 0.31 | 0.30 | 0.76 | 0.75 |
| | scFoundation | 0.72 | 0.73 | 0.78 | 0.81 | 0.54 | 0.61 | 0.78 | 0.79 | **0.99** | **0.99** | **0.97** | **0.97** | 0.88 | 0.88 | **0.93** | **0.93** |

## 4.4 TEST ON ZERO-SHOT CROSS-SPECIES KNOWLEDGE TRANSFER

**Experiment setup and evaluation metrics** We further evaluate the generalization ability of scREBOUND across species through the task of zero-shot cross-species knowledge transfer. Without finetuning the models on mouse data, we directly apply the models that is solely trained on human dataset to 200k mouse data sampled from CELLxGENE mouse cell collection (Ver. 2024-07-01). We then evaluate the quality of the cell representation of the mouse data using metrics that are also used in batch effect removal (ARI, NMI, and ASW). These metrics are selected as they are direct indicator for the information preservation and batch effect removal of cell embedding. To handle the gene set inconsistency across species, we map the mouse gene set to human gene set using gene orthologs information.

Table 5: Scores on mouse data knowledge transfer. Numbers in bold denote the best, and with underlines denote the second best.

| Methods | Mouse Data | | |
|---|---|---|---|
| | ARI | NMI | ASW |
| scREBOUND | **0.53** | 0.65 | 0.55 |
| scGPT | 0.27 | 0.44 | 0.46 |
| scMulan | 0.35 | 0.57 | 0.54 |
| UCE | 0.41 | **0.67** | 0.55 |
| scFoundation | **0.53** | 0.65 | **0.58** |

**Results** The scores are shown in Table 5. scREBOUND and scFoundation show the top performance across all methods. However, scFoundation is the slowest method out of all benchmarked methods while scREBOUND is computationally most efficient.

## 4.5 TEST ON ZERO-SHOT IMPUTATION OF MISSING GENE EXPRESSION

**Experiment setup and evaluation metrics** We randomly sub-sampled 2000 cells from each test dataset to test missing value imputation. We first applied random masking on the gene expression data (masking percentage 10% and 20%), then fed the masked gene expression data into scREBOUND and baseline models, and use their masked-gene predictor module to impute the missing expression values. We then measured the imputation accuracy by comparing the imputed values with the masked ground truth expression values.

Among the baseline scFMs, only scGPT and scFoundation provide imputation functionality. We have also included scVI (Lopez et al., 2018), which is a non-foundation model based method. For scVI, we used the remaining cells (removed of the 2000 test cells) from each test dataset to train the model for each test dataset separately. We then evaluated the scVI imputation performance by running each trained scVI on the 2000 cells of the corresponding test dataset.

To evaluate whether predicted missing values preserve the relative order of genes' expression levels in cells, we calculated Pearson and Spearman correlation between the imputed gene expression values and the masked ground truth gene expression values (denoted by PCorr and SCorr, respectively). Higher correlation scores mean better imputation results.

**Results**   The scores are shown in Table 6. Both models of scGPT do not impute meaningful gene count values, which aligns with the recent benchmark result (Liu et al., 2023). scFoundation, being the largest model with ∼100 million parameters, has the comparable or slightly better performance than scREBOUND, which has ∼20 million parameters. scVI, being a top-performing conventional method, while trained on the held-out data of each test dataset (which makes it no longer zero-shot), does not surpass the performance of scFoundation and scREBOUND.

Table 6: Imputation accuracy of scREBOUND and baseline methods under data with mask percentage 10% and 20%. Numbers in bold denote the best, and with underlines denote the second best.

| Mask % | Methods | Immune | | Pancreas | | Lung | | Covid19 | | Cell Line | | PBMC 1 | | Pancrm | | PBMC 2 | |
|---|---|---|---|---|---|---|---|---|---|---|---|---|---|---|---|---|---|
| | | SCorr | PCorr | SCorr | PCorr | SCorr | PCorr | SCorr | PCorr | SCorr | PCorr | SCorr | PCorr | SCorr | PCorr | SCorr | PCorr |
| 10% | scREBOUND | **0.37** | **0.90** | **0.43** | 0.42 | 0.35 | 0.56 | **0.37** | **0.79** | 0.51 | **0.83** | **0.38** | **0.92** | **0.46** | 0.46 | 0.27 | **0.83** |
| | scGPT-whole-human | -0.17 | -0.21 | -0.29 | -0.18 | 0.0 | 0.0 | 0.0 | 0.0 | 0.25 | 0.13 | -0.13 | -0.11 | 0.02 | -0.001 | -0.08 | -0.07 |
| | scGPT-continual | -0.18 | -0.20 | -0.06 | -0.02 | 0.0 | 0.0 | 0.0 | 0.0 | 0.06 | 0.07 | -0.28 | -0.22 | -0.08 | -0.06 | -0.12 | -0.11 |
| | scFoundation | 0.36 | 0.85 | 0.42 | **0.60** | **0.38** | **0.69** | 0.36 | 0.76 | 0.48 | 0.77 | 0.37 | 0.86 | 0.44 | **0.61** | 0.26 | **0.83** |
| | scVI | **0.37** | 0.87 | 0.39 | 0.51 | 0.32 | 0.50 | 0.35 | 0.75 | **0.52** | 0.73 | 0.35 | 0.79 | 0.41 | 0.56 | **0.30** | 0.59 |
| 20% | scREBOUND | 0.35 | 0.84 | 0.40 | 0.34 | 0.35 | 0.56 | 0.35 | 0.74 | 0.48 | **0.79** | 0.36 | **0.88** | 0.43 | 0.38 | 0.25 | 0.78 |
| | scGPT-whole-human | 0.16 | 0.13 | 0.18 | 0.17 | 0.02 | 0.05 | 0.10 | 0.06 | 0.07 | 0.01 | -0.01 | -0.05 | -0.01 | -0.05 | -0.02 | -0.05 |
| | scGPT-continual | 0.06 | 0.02 | 0.18 | 0.14 | 0.02 | 0.02 | 0.10 | 0.05 | -0.28 | -0.25 | -0.26 | -0.17 | 0.02 | 0.02 | -0.14 | -0.14 |
| | scFoundation | **0.36** | 0.84 | **0.42** | 0.53 | **0.38** | **0.69** | **0.36** | 0.74 | 0.48 | 0.77 | **0.37** | 0.85 | **0.44** | 0.55 | 0.26 | **0.82** |
| | scVI | **0.36** | **0.87** | 0.39 | **0.57** | 0.31 | 0.49 | 0.35 | **0.75** | **0.51** | 0.71 | 0.35 | 0.77 | 0.41 | 0.54 | **0.29** | 0.57 |

## 4.6 ABLATION STUDY

We conducted ablation tests on the use of multi-granular contrastive regularization and batch encoder. We tested three versions of scREBOUND: (1) Original model ("Batch Enc & Contr" model); (2) Model trained without contrastive regularization but with batch encoder ("Batch Enc" model); (3) Model trained without batch encoding and contrastive regularization ("Vanilla" model). We evaluated the models' MTP ability through the task of missing expression imputation, and the models' ability to handle batch effect through the tasks of batch effect removal and cell type annotation (test setting and evaluation metrics follow Secs. 4.2, 4.3, 4.5). We selected four larger datasets with more experimental batches for the evaluation ("Immune", "Pancreas", "Lung", "Covid19").

The test result of expression imputation is shown in Appendix Table 8. The results of batch effect removal and cell type annotation are shown in Appendix Fig. 4 and Fig. 5. Under the task of expression imputation, "Batch Enc" model shows consistently better performance compared to "Vanilla" model, and "Batch Enc & Contr" has top performance among all three models, which proves the effectiveness of batch encoding and contrastive regularization in improving model imputation ability. Under the task of batch effect removal and cell type annotation, model with batch encoding ("Batch Enc") shows consistently better performance compared to "Vanilla" model on three out of four test datasets (except "Immune"), and the use of contrastive regularization ("Batch Enc & Contr") significantly improves the model performance among all four datasets.

## 5 CONCLUSION AND FUTURE WORK

The primary strength of scREBOUND lies in its computational efficiency. scREBOUND reduces inference time by at least an order of magnitude compared to existing scFMs, making it feasible for deployment in wet-lab and clinical environments, where rapid and reliable single-cell analysis is critical. While being highly efficient, scREBOUND also achieves performance that matches or exceeds that of current scFMs on downstream tasks, including batch effect removal, cell type classification, and missing gene expression imputation. This is made possible through its biology-informed feature compression, multi-granular contrastive regularization, and novel batch representation learning. Ablation studies confirm that both the batch encoder and the contrastive loss design contribute significantly to the model's ability to address batch effects. The effectiveness of scREBOUND highlights the potential for building compact and high-performing scFMs.

As the first "small" scFM, scREBOUND has certain limitations. Currently, it is trained on healthy human scRNA-seq data and has not yet been extended to broader settings, such as multi-species, multi-condition, or multi-modal single-cell data. Additionally, it has not been applied to some biological tasks such as gene regulatory network inference. In future work, we plan to extend scREBOUND into a cross-species, cross-domain single-cell foundation model with continued emphasis on efficient design, and to broaden its applications to a wider range of downstream tasks.

## 6 REPRODUCIBILITY STATEMENT

To ensure the reproducibility of the model, the link to the complete source code of scREBOUND is provided in Appendix Sec. A.9. The sources and download links of the training and evaluation datasets are also provided in Appendix Sec. A.9.

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

## A  APPENDIX

### A.1  FURTHER DETAILS ON MODEL DESIGN OF RELATED WORK

UCE (Rosen et al., 2023) constructs a cell sentence by sampling a bag of 1024 genes with replacement from each cell, with sampling probability for each gene proportional to their expression levels. It then imposes a pseudo-ordering of genes based on their genomic locations. During training, it randomly masks 20% of genes from the bag sampling, and uses a binary predictor to predict whether a masked gene is expressed or not (Lin et al., 2022). Geneformer (Theodoris et al., 2023) ranks genes by expression level and uses this ranking as the token order. It masks 15% of tokens and trains the model to predict the identities of the masked genes. scGPT (Cui et al., 2024) constructs gene tokens that encode both identity and expression of the gene. During training, it randomly masked the expression of genes and predict the masked expression following a attention-score-based ordering. scMulan (Bian et al., 2024b) constructs cell sentences with task token, meta-data token along with gene tokens, and re-orders the tokens into "Q&A" sentence format for pretraining.

### A.2  scREBOUND MODEL ARCHITECTURE

The overall architecture of scREBOUND follows Fig. 1b. scREBOUND uses a 6-layer transformer with 8 attention heads, each head having 64 dimensions, to encode the input. For predicting masks, it uses a separate 6-layer MLP. The final cell representations are stored in a 256-dimensional space.

### A.3  THEORETICAL ANALYSIS ON THE EFFICIENCY OF scREBOUND COMPRESSION STRATEGY

The gene compression unit in scREBOUND based on the mincut loss upper-bounds the mutual information loss after the feature compression. We first define the feature vectors (gene token embedding) as $\mathbf{X} \in \mathbb{R}^{n \times d}$ ($n$: number of genes, $d$: number of latent dimensions), and the corresponding gene-gene association graph as $G(\mathbf{X}, W)$ where $w_{uv}$ is the edge weight between features $u$ and $v$. The gene compression framework learns the compression matrix $\mathbf{S} \in \mathbb{R}^{n \times m}$ ($m$: number of meta-genes), and compress the feature vectors into $\mathbf{Z} = \mathbf{S}^T \mathbf{X}$. We further define the set of genes assigned to a metagene $i$ as $C_i = \{u | S_{u,i} \neq 0\}$, and the mincut loss follows

$$L_c = \sum_{i \neq j} \sum_{u \in C_i, v \in C_j} w_{uv} \tag{9}$$

**Theorem 1.** *If the edge weights $w_{uv}$ in graph $G$ follows Gaussian similarity, i.e. $w_{uv} = \exp\left(-\frac{1}{2\sigma^2} \|\mathbf{X}_u - \mathbf{X}_v\|_2^2\right)$. Then the mutual information loss $H(\mathbf{X}) - I(\mathbf{Z}; \mathbf{X})$ is upper-bounded by the mincut loss used in the compression unit.*

*Proof.* The mutual information loss follows

$$H(\mathbf{X}) - I(\mathbf{Z}; \mathbf{X}) = H(\mathbf{X}|\mathbf{Z}) \tag{10}$$

Minimizing the mutual information loss is then equivalent to minimizing the conditional entropy $H(\mathbf{X}|\mathbf{Z})$. Given totally $m$ meta-genes, $H(\mathbf{X}|\mathbf{Z})$ can be re-written as

$$H(\mathbf{X}|\mathbf{Z}) = \sum_{i=1}^{m} P(\mathbf{Z}_i) H(\mathbf{X}|\mathbf{Z}_i) \tag{11}$$

If we assume that $\forall u \in C_i$, the node feature $\mathbf{X}_u$ follows Gaussian distribution $\mathcal{N}(\mathbf{Z}_i, \mathbf{\Sigma}_i)$, then $H(\mathbf{X}|\mathbf{Z}_i)$ can be rewritten as

$$H(\mathbf{X}|\mathbf{Z}_i) = 0.5 \log((2\pi e)^d \det(\mathbf{\Sigma}_i)) \tag{12}$$

$H(\mathbf{X}|\mathbf{Z}_i)$ is then minimized by minimizing $\det(\mathbf{\Sigma}_i)$ for $i$. The within-cluster variance can be calculated as

$$\det(\mathbf{\Sigma}_i) = \sum_{u \in C_i} \|\mathbf{X}_u - \mathbf{Z}_i\|^2 \tag{13}$$

Assuming the weight between gene $u$ and $v$, $w_{uv}$, follows Gaussian distribution:

$$1 - w_{uv} = 1 - \exp\left(-\frac{1}{2\sigma^2}\|\mathbf{X}_u - \mathbf{X}_v\|_2^2\right) \geq \frac{1}{2\sigma^2}\|\mathbf{X}_u - \mathbf{X}_v\|_2^2 \tag{14}$$

And we have

$$\|\mathbf{X}_u - \mathbf{X}_v\|_2^2 \leq 2\sigma^2(1 - w_{uv}) \tag{15}$$

Using the centroid identity, we have

$$
\begin{aligned}
&\sum_{u \in C_i} \|\mathbf{X}_u - \mathbf{Z}_i\|_2^2 \\
&\leq \sigma^2\left(|C_i| - \frac{1}{|C_i|}\sum_{u,v \in C_i} w_{uv}\right) \\
&= \sigma^2\left(|C_i| - \frac{1}{|C_i|}(\sum_{u,v} w_{uv} - L_c)\right)
\end{aligned}
\tag{16}
$$

The variance within each cluster is upper-bounded by the mincut loss $L_c$. Following Equ. 12, the mutual information loss is also upper-bounded by $L_c$. The theorem is proved. $\square$

### A.4 Construction of cell type bin-code from cell ontology tree

The label bin-code is an $k$ dimensional vector where $k$ is the total number of nodes/labels in the cell ontology tree. Each bin-code element within the vector corresponds to one node in the tree with value either $0$ or $1$. Given a cell type label, we first identify its corresponding node and find all its ancestral nodes in the cell ontology tree. We set the bin-code elements of the node and its ancestral nodes to 1, and the remaining elements to 0.

For each anchor cell $i$, we then segregate the target cells into the positive pool $\mathbb{P}(i)$, negative pool $\mathbb{N}(i)$, and mutual pool $\mathbb{M}(i)$ by comparing their bin-codes with the bin-code of the anchor cell:

- Positive pool $\mathbb{P}(i)$ contains target cells for which the conditions holds: For every "1" in the bin-code of the anchor cell, the corresponding position in the bin-code of the target cell also shows "1".

- Neutral pool $\mathbb{M}(i)$ contains target cells for which the condition holds: If there is a "1" in the target cell bin-code, the corresponding position in the anchor cell is also "1".

- Negative pool $\mathbb{N}(i)$ contains the target cells not present in both $\mathbb{P}(i)$ and $\mathbb{M}(i)$.

### A.5 Construction of batch encoder

In scREBOUND, we construct a batch encoder to learn batch representation from batch-related information. We curate batch-specific features from the gene expression values for each training batch. The full list of features that we collect is included in Table 7.

Table 7: Summary of features used for batch encoder

| Feature Name | # Features | # Categories |
|---|---|---|
| mean proportion of non-zero genes | 1 | 10 |
| mean expression of non-zero genes | 1 | 10 |
| mean proportion of mito-gene | 1 | 10 |
| mean expression of house keeping genes | 19 | 10 |
| mean expression of ribosomal genes | 11 | 10 |
| mean expression of stress-related genes | 6 | 10 |

Mean proportion of non-zero genes measures the average proportion of genes with non-zero expression (non-zero genes) for each batch. To compute the feature of batch $b$, we first calculate the

proportion of non-zero genes for each cell belonging to $b$, and then average the proportion values across these cells.

Mean expression of non-zero genes measures the average expression values of the non-zero genes for each batch. To compute the feature of batch $b$, we first calculate the mean expression value of non-zero genes for each cell in $b$, and then average these mean expression value across these cells.

Mean proportion of mito-gene measures the average proportion of mitochondrial gene counts for each batch. Cells with higher mito-gene counts are usually stressed or damaged, which affect the original gene expression distribution of the cell (Luecken & Theis, 2019). As a result, mean proportion of mito-gene count is a good indicator of batch effect. For batch $b$, we first measure the proportion of mito-gene counts (among all counts) of each cell in $b$, and then average the proportion across these cells.

Mean expression of house keeping genes: we incorporate 19 house keeping genes (Hounkpe et al., 2021). House keeping genes are genes that consistently expressed across cell types, tissues, and experimental conditions. The expression of these genes can be used as baseline expression for the comparison of batch effects. For each batch, we concatenate the average expressions of these house keeping genes as the batch-specific features.

Mean expression of ribosomal genes: we also incorporate the batch-specific average expressions of 11 ribosomal genes, since the expression of these genes are sensitive to batch effect. The ribosomal genes are selected from gene list in Gene Ontology term GO: 0005840 (Ashburner et al., 2000) with the highest mean expression variation across batches.

Mean expression of stress-related genes: we select 6 stress genes that are also sensitive to batch effect and use their average expressions in each batch as batch-specific features. The stress gene are selected from Gene Ontology term GO: 0006950 (Ashburner et al., 2000) with the highest mean expression variation across batches.

For each batch, we then digitize the values of each feature into 10 bins, and feed the digitized value into the batch encoder. The batch encoder learns a 256-dimensional embedding for each of the 10 bins corresponding to every feature. To generate the embedding of batch $b$, the encoder then concatenates the bin embedding of all $b$'s features and feed it to a 2-layer MLP.

## A.6 FURTHER DETAILS ON MODEL TRAINING

Before the training of scREBOUND starts, we initialize the meta-gene assignment matrix $\mathbf{S}$ by minimizing Equ. 1 through gradient descent:

$$\mathbf{S}^{t+1} = \mathbf{S}^t - \eta \nabla_{\mathbf{S}} \mathcal{L}_G(\mathbf{S}, \mathbf{G}) \tag{17}$$

$\mathbf{S}^0$ is set using the spectral clustering result of $\mathbf{G}$ (number of clusters 256), and $\eta$ is the learning rate ($\eta = 0.01$).

scREBOUND is then trained with 4 Nvidia A40 GPUs on 35 million CELLxGENE human healthy dataset for 3 epochs, where training **stage 1**, **stage 2**, **stage 3** each takes up 1 epoch (Sec. 3.5). We used AdamW as the training optimizer and schedule the learning rate with one-cycle learning rate policy. The training batch size is set to 2048, and the maximum learning rate of the model is set to $2.4 \times 10^{-5}$.

## A.7 DETAILS ON EVALUATION METRICS

### A.7.1 ARI, NMI AND ASW SCORES

Given the clustering output and ground truth cell type label, ARI and NMI scores measure the how similar the cluster label is to the ground truth label. Both scores range between 0 and 1, where a higher score means a better batch integration performance.

On the other hand, ASW score directly measure the separation of cell types in the embedding space by comparing the distance between cells of different cell types with the distance between cells of the same cell type. The ASW score ranges between -1 and 1. A higher score means a better separation of cells of different cell types and a better mixing of cells of the same cell type.

**ARI**   Given the ground truth cell type label and the cluster label, we first construct the contingency table $\mathbf{N}$, where element $n_{ij}$ encodes the number of cells with both ground truth label $i$ and cluster label $j$. ARI score then follows:

$$\text{ARI} = \frac{\sum_{ij} \binom{n_{ij}}{2} - \left[\sum_i \binom{a_i}{2} \sum_j \binom{b_j}{2} \Big/ \binom{n}{2}\right]}{\frac{1}{2} \left[\sum_i \binom{a_i}{2} + \sum_j \binom{b_j}{2}\right] - \left[\sum_i \binom{a_i}{2} \sum_j \binom{b_j}{2} \Big/ \binom{n}{2}\right]} \tag{18}$$

$a_i$ is the row sum of the contingency table: $a_i = \sum_j n_{ij}$. $b_j$ is the column sum of the contingency table: $b_j = \sum_i n_{ij}$. $n$ is the total number of cells.

**NMI**   Given the ground truth cell type label $g = \{g_1, g_2, \cdots, g_n\}$ and the cluster label $c = \{c_1, c_2, \cdots, c_n\}$, NMI score follows:

$$\text{NMI}(g, c) = \frac{2I(g; c)}{H(g) + H(c)} \tag{19}$$

$I(g; c)$ is the mutual information between ground truth label and cluster label. $H(g)$ is the entropy of the ground truth label, and $H(c)$ is the entropy of cluster label.

**Calculate ARI and NMI with optimal clustering**   We calculate ARI and NMI scores following the scIB benchmarking pipeline (Luecken et al., 2022). We cluster the cells by applying Leiden clustering algorithm (Traag et al., 2019b) on cell embeddings. To decide the optimal cluster "resolution" for the Leiden clustering algorithm, we conduct grid search on the "resolution" starting from 0 to 2 with step-size 0.2 and select the optimal clustering result that has the highest NMI score against the ground truth label. For the cell embedding generated by each method, we conduct the grid search and optimal clustering separately, and then calculate the ARI and NMI score between the optimal clustering label and the ground truth cell type label.

**ASW**   ASW is calculated directly on cell embedding using the ground truth cell type label. For each cell $i$, the silhouette width (Rousseeuw, 1987) measures the average distance of the cell to other cells of the same cell type $d_1(i)$, and the average distance of the cell to the other cells of the closest cell type $d_2(i)$, and then calculates score following:

$$s(i) = \frac{d_2(i) - d_1(i)}{\max(d_1(i), d_2(i))} \tag{20}$$

ASW is then obtained by averaging silhouette scores across all cells.

### A.7.2   WEIGHTED F1-SCORE

Given the predicted and ground truth cell type labels, the weighted F1 score is calculated following:

$$\text{F1}_{weighted} = \frac{\sum_i n_i \cdot \text{F1}_i}{\sum_i n_i} \tag{21}$$

$n_i$ is the number of cells of cell type $i$. $\text{F1}_i$ is the F1 score of cell type $i$ that is calculated using the predicted and ground truth label of cell type $i$, using the formula:

$$\text{F1}_i = \frac{\text{TP}}{(\text{TP} + 0.5 * (\text{FP} + \text{FN}))} \tag{22}$$

where TP is true positive, FP is false positive and FN is false negative. Weighted F1-score ranges between 0 and 1, where a higher score means a better classification performance.

## A.8 ABLATION TEST RESULTS

Table 8: Ablation test on data imputation with mask percentage 10% and 20%. Numbers in bold denote the best, and with underlines denote the second best.

| Mask % | Methods | Immune | | Pancreas | | Lung | | Covid19 | |
|---|---|---|---|---|---|---|---|---|---|
| | | SCorr | PCorr | SCorr | PCorr | SCorr | PCorr | SCorr | PCorr |
| 10% | Vanilla | 0.364 | 0.878 | 0.422 | 0.384 | 0.371 | **0.604** | 0.366 | 0.778 |
| | Batch Enc | 0.369 | 0.896 | 0.428 | 0.407 | 0.372 | 0.592 | 0.368 | 0.786 |
| | Batch Enc & Contr | **0.370** | **0.900** | **0.431** | **0.425** | **0.373** | 0.602 | **0.369** | **0.787** |
| 20% | Vanilla | 0.343 | 0.828 | 0.393 | 0.313 | 0.349 | **0.560** | 0.345 | 0.740 |
| | Batch Enc | 0.348 | 0.843 | 0.400 | 0.333 | **0.351** | 0.548 | 0.347 | **0.743** |
| | Batch Enc & Contr | **0.350** | **0.844** | **0.402** | **0.346** | 0.351 | 0.559 | **0.348** | 0.741 |

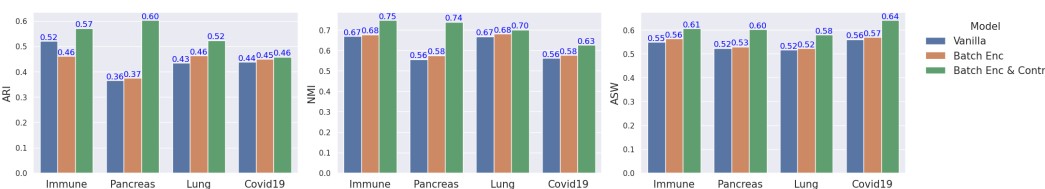

Figure 4: Barplot result of ablation test on batch effect removal.

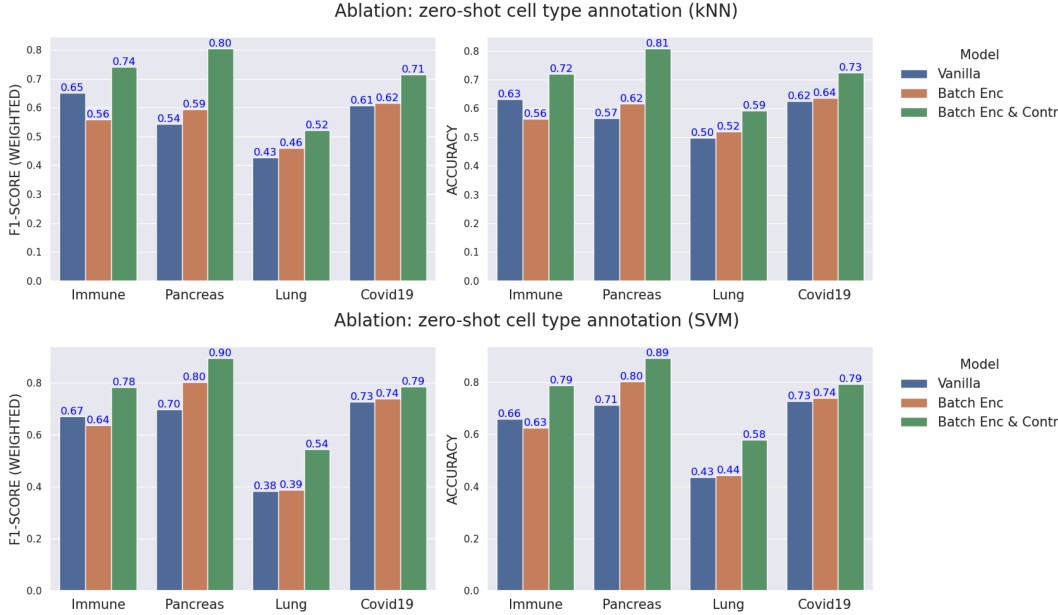

Figure 5: Barplot result of ablation test on cell type annotation.

### A.9 DATA AND CODE AVAILABILITY

The CELLxGENE atlas for pretraining can be found on its website: `https://cellxgene.cziscience.com/collections` and download through python package cellxgene-census. The test datasets "Immune", "Pancreas", "Lung" can be downloaded through link `https://figshare.com/articles/dataset/Benchmarking_atlas-level_data_integration_in_single-cell_genomics_-_integration_task_datasets_Immune_and_pancreas_/12420968`. The test dataset "Covid19" can be downloaded from Gene Expression Omnibus with accession number GSE155673, GSE149689, and GSE150728. The sources and download links of the 4 scEval datasets ("CellLine", "PBMC1", "Pancrm", "PBMC2") are provided in the supplementary material of the original manuscript (Li et al., 2025).

The scREBOUND package is available at `https://anonymous.4open.science/r/scREBOUND-3C3B`, and the full evaluation scripts are available at `https://anonymous.4open.science/r/scREBOUND_test-2189`.

### A.10 VISUALIZATIONS OF CELL REPRESENTATION

We visualized the cell embedding of scREBOUND and baseline methods using UMAP, and annotated the cells by the ground truth cell type label and cells' batch id (Figs. 6, 7). Ideally, cell embedding that is removed of batch effect should have (1) cells of same cell type aligned across batches in the embedding space, and (2) cells of different cell types well separated in the embedding space.

Fig. 6 shows the cell embedding visualization of scREBOUND and baseline methods on "Immune" dataset. Among all methods, scREBOUND and scGPT have the best batch alignment and cell type separation in the cell embedding space. scREBOUND performs better than scGPT especially for the alignment of CD4+ T cells and CD14+ Monocytes (blue and orange cells in the left column of Fig. 6) across batches.

Fig. 7 shows the cell embedding visualization of "Pancreas" dataset. Among all methods, scREBOUND and scGPT have the best batch alignment and cell type separation in the cell embedding space. However, scREBOUND successfully segregate the delta and gamma (dark green and purple cells in the left column of Fig. 7) cells, whereas scGPT cannot distinguish these cell types from alpha and beta cells (yellow and orange cells in the left column of Fig. 7).

### A.11 EXPERIMENTS COMPUTE RESOURCES

Pretraining of scREBOUND was done using four Nvidia A40 GPUs with 48 GB of GDDR6 memory. Inference of each test dataset was performed using one Nvidia A40 GPU with 48 GB of GDDR6 memory.

### A.12 POTENTIAL SOCIETAL IMPACTS

scREBOUND's efficient design democratizes access to single-cell analysis, and indirectly contributes to healthcare and medicine by enabling more labs to contribute to precision medicine. However, we caution that biased training data could amplify health disparities, necessitating careful dataset curation. We emphasize the importance of using balanced datasets.

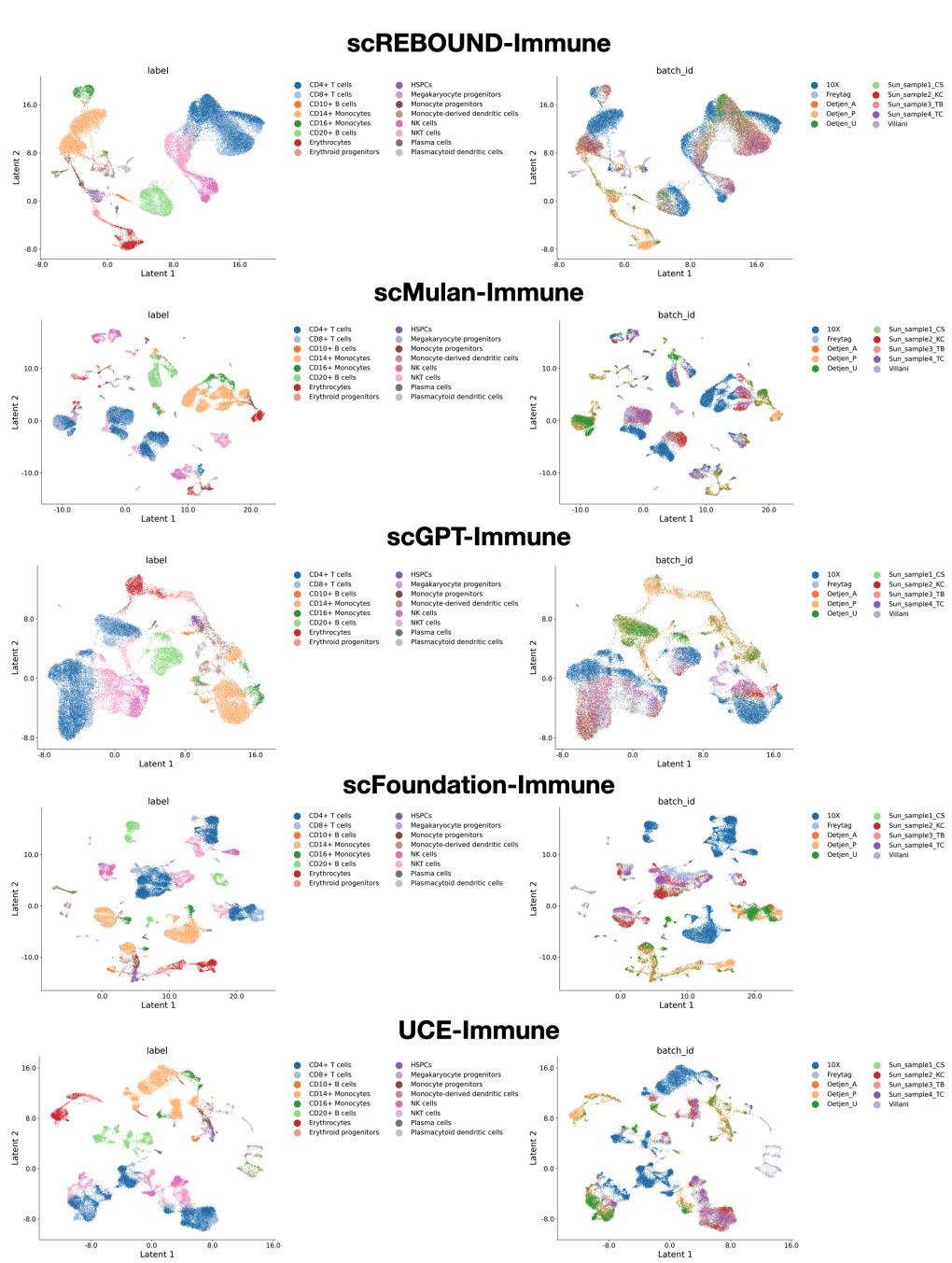

Figure 6: UMAP visualization of "Immune" dataset cell embedding from scREBOUND and baseline methods (scMulan, scGPT, scFoundation, and UCE). Cells are colored by the ground truth cell type label in the left column and the batch id in the right column.

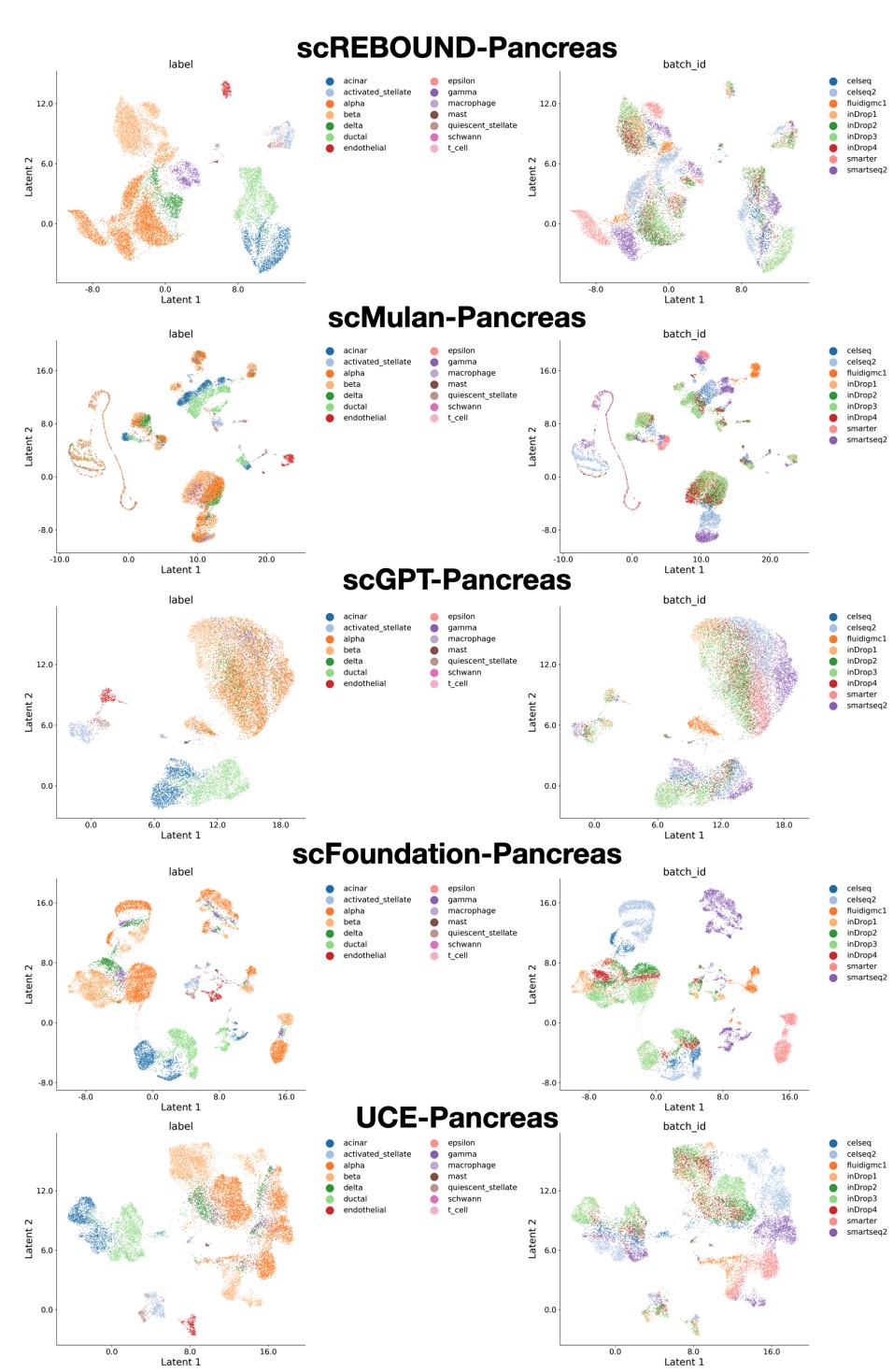

Figure 7: UMAP visualization of "Pancreas" dataset cell embedding from scREBOUND and baseline methods (scMulan, scGPT, scFoundation, and UCE). Cells are colored by the ground truth cell type label in the left column and the batch id in the right column.

