# OpenReview forum: "scREBOUND: An Efficient Design of single-cell Foundation Model with Batch Representation"
_ICLR.cc/2026/Conference — Submitted to ICLR 2026_

### Official Review · Reviewer_GCU6 · 2025-10-31

**Soundness:** 3
**Presentation:** 3
**Contribution:** 2
**Rating:** 6
**Confidence:** 3

**Summary:**

The authors propose an encoder only scRNA-seq foundation model, by compressing genes into clusters or meta-genes using ESM2-guided clustering .The training process follows a multi-granular contrastive loss with the addition of a min-cut regularizer and a batch encoder.
Overall, the model shows promising performance, particularly being more efficient than other baselines (e.g. scFoundation, scGPT) on batch integration, annotation, cross-species transferand imputation.

**Strengths:**

- It's noteworthy that the model is quite efficient: lower inference time and memory cost vs SCGPT

- The approach on the ESM2-guided clustering seems natural. It gives a biology-aware compression notion, that is quite crucial fighting for efficiency in scFMs. I find it principled and easy to reason about it.

- I like how the paper handles mismatched label depths across batches through the bin-code contrastive setup.

- overall, the results seem solid across integration, annotation, cross-species trasfner, and imputation
- I find the methodolgy clearly written and easy to follow up.

**Weaknesses:**

- I observe that there is a dependence on labels/ontologies. What happens with noisy or missing ontology mappings?
- I find the sensitivity of the methodology being limited explored. Particularly, the authors propose a fixed 256 meta-gene parameter. Why 256? It'd be very useful to see the performance behavior with respect to varying sizes of clusters. How robust is the performance vs tissue-specfiic settings?

**Questions:**

- Check the Weaknesses section.
- Moreover, the paper focuses on zero-shot settings. How would scREBOUND + finetuning compare to scGPT or scFoundation when they're finetuned as well?
- Regarding systemic mislabels, does the neutral pool mask over them? or does the performance degrade?

---

> ### Author Response · Authors · 2025-12-03
>
> ```
> Q1. I observe that there is a dependence on labels/ontologies. What happens with noisy or missing ontology mappings?
> ```
> **A1**.
> We agree that the moisy or missing cell type ontology mapping in the training dataset could potentially affect the model performance. According to the reviewer's suggestion, we conduct experiments evaluating how robust the model is towards the labeling noise (cell ontoly mapping that are missing & erroneous). We corrupted the information of 5%, 10% and 20% of the labels in the cell type ontology tree:
> * Firstly, we randomly select x% (x = 2.5, 5, 10) of the labels within the cell ontology tree, and remove all the connections between the selected label and the remaining labels (create "stand-alone" label).
> * Then, we further select x% (x = 2.5, 5, 10) of the labels within the cell ontology tree, and randomly add false positive or remove true positive connections between the selected label and the remaining labels (create "wrongly-annotated" labels).
>
> We then train scREBOUND with corrupted annotation, and evaluate its performance on (1) batch effect removal, (2) cell type annotation, and (3) data imputation. The results (Tables 1~3) show that scREBOUND is robust to the label corruption.
>
>
> **Table 1**. Batch effect removal scores with different label noise corruption
> |label corruption|Immune|||||Pancreas|||||Lung|||||Covid19|||||PBMC 1|||||Pancrm|||||PBMC 2|||||
> |-|-|-|-|-|-|-|-|-|-|-|-|-|-|-|-|-|-|-|-|-|-|-|-|-|-|-|-|-|-|-|-|-|-|-|-|
> ||ARI|NMI|ASW|ASW (batch)|Graph Connectivity|ARI|NMI|ASW|ASW (batch)|Graph Connectivity|ARI|NMI|ASW|ASW (batch)|Graph Connectivity|ARI|NMI|ASW|ASW (batch)|Graph Connectivity|ARI|NMI|ASW|ASW (batch)|Graph Connectivity|ARI|NMI|ASW|ASW (batch)|Graph Connectivity|ARI|NMI|ASW|ASW (batch)|Graph Connectivity|
> |0%|0.57|0.75|0.61|0.86|0.97|0.60|0.74|0.60|0.86|0.89|0.52|0.70|0.58|0.88|0.97|0.46|0.63|0.64|0.85|0.97|0.57|0.69|0.64|0.84|0.90|0.66|0.75|0.60|0.84|0.90|0.81|0.79|0.68|0.83|0.95|
> |5%|0.50|0.74|0.62|0.86|0.96|0.50|0.70|0.58|0.87|0.88|0.52|0.70|0.58|0.88|0.97|0.46|0.62|0.64|0.85|0.97|0.55|0.69|0.66|0.83|0.87|0.59|0.73|0.58|0.85|0.90|0.80|0.78|0.68|0.81|0.95|
> |10%|0.58|0.75|0.62|0.86|0.96|0.58|0.72|0.58|0.86|0.87|0.51|0.70|0.58|0.87|0.97|0.45|0.62|0.64|0.85|0.97|0.58|0.70|0.66|0.83|0.90|0.64|0.74|0.58|0.85|0.93|0.79|0.78|0.68|0.83|0.95|
> |20%|0.57|0.74|0.61|0.86|0.96|0.55|0.70|0.57|0.87|0.90|0.52|0.70|0.58|0.88|0.97|0.45|0.61|0.63|0.85|0.97|0.55|0.68|0.65|0.85|0.89|0.55|0.71|0.58|0.84|0.92|0.78|0.77|0.67|0.83|0.95|
>
>
> **Table 2**. Cell type annotation accuracy of the models with different label noise corruptions
> |Classifier|label corruption|Immune||Pancreas||Lung||Covid19||PBMC 1||Pancrm||PBMC 2||
> |-|-|-|-|-|-|-|-|-|-|-|-|-|-|-|-|
> |||WF1|Accu|WF1|Accu|WF1|Accu|WF1|Accu|WF1|Accu|WF1|Accu|WF1|Accu|
> |knn|0%|0.74|0.72|0.80|0.80|0.52|0.59|0.71|0.72|0.72|0.72|0.72|0.72|0.50|0.49|
> ||5%|0.69|0.67|0.79|0.80|0.52|0.60|0.72|0.73|0.74|0.74|0.68|0.69|0.49|0.49|
> ||10%|0.67|0.66|0.79|0.80|0.52|0.60|0.72|0.73|0.75|0.75|0.74|0.75|0.45|0.47|
> ||20%|0.67|0.65|0.81|0.82|0.48|0.56|0.74|0.74|0.75|0.75|0.75|0.76|0.46|0.47|
> |svm|0%|0.78|0.79|0.90|0.90|0.55|0.58|0.78|0.79|0.81|0.81|0.87|0.87|0.59|0.59|
> ||5%|0.74|0.73|0.79|0.80|0.52|0.57|0.79|0.80|0.83|0.83|0.85|0.84|0.60|0.59|
> ||10%|0.72|0.71|0.88|0.87|0.52|0.57|0.78|0.79|0.80|0.81|0.88|0.87|0.52|0.54|
> ||20%|0.73|0.72|0.87|0.87|0.47|0.52|0.80|0.80|0.84|0.84|0.86|0.86|0.56|0.57|
>
> **Table 3**. Data imputation of the models with different label noise corruptions
> |Mask|label corruption|Immune||Pancreas||Lung||Covid19||CL||PBMC 1||Pancrm||PBMC 2||
> |-|-|-|-|-|-|-|-|-|-|-|-|-|-|-|-|-|-|
> |||Pcorr|Scorr|Pcorr|Scorr|Pcorr|Scorr|Pcorr|Scorr|Pcorr|Scorr|Pcorr|Scorr|Pcorr|Scorr|Pcorr|Scorr|
> |10%|0%|0.90|0.37|0.43|0.43|0.61|0.37|0.79|0.37|0.84|0.51|0.92|0.38|0.46|0.46|0.83|0.27|
> ||5%|0.90|0.37|0.42|0.43|0.61|0.37|0.78|0.37|0.84|0.51|0.92|0.38|0.44|0.46|0.83|0.27|
> ||10%|0.90|0.37|0.43|0.43|0.61|0.37|0.78|0.37|0.84|0.51|0.92|0.38|0.45|0.46|0.83|0.27|
> ||20%|0.90|0.37|0.42|0.43|0.60|0.37|0.78|0.37|0.84|0.51|0.92|0.38|0.45|0.46|0.83|0.27|
> |20%|0%|0.89|0.37|0.37|0.42|0.59|0.37|0.78|0.37|0.83|0.51|0.92|0.38|0.40|0.45|0.83|0.27|
> ||5%|0.89|0.37|0.36|0.42|0.59|0.37|0.78|0.37|0.83|0.51|0.92|0.38|0.38|0.45|0.83|0.27|
> ||10%|0.89|0.37|0.37|0.42|0.59|0.37|0.78|0.37|0.83|0.51|0.92|0.38|0.39|0.45|0.83|0.27|
> ||20%|0.89|0.37|0.36|0.42|0.59|0.37|0.78|0.37|0.83|0.51|0.92|0.38|0.38|0.45|0.83|0.27|

---

> ### Author Response · Authors · 2025-12-03
>
> ```
> Q2. I find the sensitivity of the methodology being limited explored. Particularly, the authors propose a fixed 256 meta-gene parameter. Why 256? It'd be very useful to see the performance behavior with respect to varying sizes of clusters. How robust is the performance vs tissue-specfiic settings?
> ```
>
> **A2**. 256 is a number that we picked considering the trade-off between the model performance and the running speed. More meta-genes will likely lead to higher precision but increase running time, but fewer meta-genes may cause too much information loss.
>
>
> According to the reviewer's comment, we conduct comprehensive hyper-parameter test on scREBOUND with 64, 128, 256, 300, and 400 meta-genes. We select 4 test datasets for the evaluation: Immune, Pancreas, Lung, and Covid19. The results are shown in Tables 4-6. We cannot conduct test with more than 400 meta-genes due to the resources and time limit. The training gets extremely slow and consumes significantly larger GPU memory with more than 400 meta-genes, since the complexity of the computation is $O(n^2)$ with the number of tokens $n$. In addition, the advantage of scREBOUND, i.e. computational efficiency, would not be significant with too many meta-genes.
>
> Model with 300 and 400 meta-genes does not show significant performance gain compared to model with 256 meta-genes, but the inference time is significantly longer than the model with 256 meta-genes (e.g. model with 300 meta-genes has 1.4X of the inference time compared to 256 meta-genes). Model with 128 meta-genes and 64 meta-genes are faster than model with 256 meta-genes, but the performance is also the lower compared to the others especially in the tasks of cell type annotation and data imputation (Tables 5-6). The result shows that the number of meta-genes is selected based on the trade-off between inference efficiency and model performance. More meta-gene leads to better model performance, but slower inference speed.
>
> **Table 4**. Batch effect removal scores with different number of meta-genes
> |# tokens|Immune|||Pancreas|||Lung|||Covid19|||
> |-|-|-|-|-|-|-|-|-|-|-|-|-|
> ||ARI|NMI|ASW|ARI|NMI|ASW|ARI|NMI|ASW|ARI|NMI|ASW|
> |400|0.62|0.77|0.62|0.48|0.67|0.56|0.49|0.71|0.58|0.45|0.63|0.65|
> |300|0.62|0.76|0.62|0.49|0.69|0.58|0.47|0.70|0.57|0.46|0.62|0.64|
> |256|0.57|0.75|0.61|0.60|0.74|0.60|0.52|0.70|0.58|0.46|0.63|0.64|
> |128|0.75|0.79|0.60|0.49|0.69|0.59|0.47|0.69|0.57|0.45|0.62|0.64|
> |64|0.77|0.78|0.62|0.63|0.73|0.59|0.47|0.69|0.55|0.45|0.61|0.63|
>
> **Table 5**. Cell type annotation scores with different number of meta-genes
> |Classifier|# tokens|Immune||Pancreas||Lung||Covid19||
> |-|-|-|-|-|-|-|-|-|-|
> |||WF1|Accu|WF1|Accu|WF1|Accu|WF1|Accu|
> |kNN|400|0.65|0.63|0.68|0.69|0.51|0.58|0.71|0.73|
> ||300|0.72|0.70|0.75|0.76|0.51|0.59|0.69|0.70|
> ||256|0.74|0.72|0.80|0.81|0.52|0.59|0.71|0.73|
> ||128|0.74|0.73|0.74|0.76|0.51|0.58|0.69|0.70|
> ||64|0.71|0.68|0.79|0.80|0.45|0.53|0.71|0.72|
> |SVM|400|0.75|0.73|0.78|0.77|0.49|0.52|0.81|0.82|
> ||300|0.78|0.77|0.83|0.82|0.51|0.54|0.80|0.81|
> ||256|0.78|0.79|0.90|0.89|0.54|0.58|0.79|0.79|
> ||128|0.74|0.73|0.81|0.80|0.51|0.55|0.79|0.80|
> ||64|0.76|0.64|0.81|0.80|0.43|0.49|0.79|0.79|
>
> **Table 6**. Data imputation
> |Mask %|# tokens|Immune||Pancreas||Lung||Covid19||
> |-|-|-|-|-|-|-|-|-|-|
> |||SCorr|PCorr|SCorr|PCorr|SCorr|PCorr|SCorr|PCorr|
> |10%|400|0.36|0.88|0.42|0.40|0.37|0.59|0.36|0.77|
> |10%|300|0.37|0.89|0.42|0.43|0.37|0.60|0.36|0.78|
> |10%|256|0.37|0.90|0.43|0.42|0.35|0.56|0.37|0.79|
> |10%|128|0.36|0.87|0.41|0.36|0.37|0.57|0.36|0.75|
> |10%|64|0.37|0.90|0.43|0.44|0.36|0.56|0.36|0.78|
> |20%|400|0.34|0.83|0.39|0.32|0.34|0.54|0.34|0.73|
> |20%|300|0.34|0.84|0.39|0.36|0.35|0.55|0.34|0.74|
> |20%|256|0.35|0.84|0.40|0.34|0.35|0.56|0.35|0.74|
> |20%|128|0.34|0.81|0.38|0.29|0.34|0.53|0.34|0.71|
> |20%|64|0.35|0.84|0.40|0.38|0.34|0.51|0.34|0.74|

---

> ### Author Response · Authors · 2025-12-03
>
> ```
> Q3. Moreover, the paper focuses on zero-shot settings. How would scREBOUND + finetuning compare to scGPT or scFoundation when they're finetuned as well?
> Regarding systemic mislabels, does the neutral pool mask over them? or does the performance degrade?
> ```
> **A3**. We agree with the reviewer that light fine-tuning is widely used in many practical applications, in addition to directly applying foundation models in a zero-shot manner. Following the reviewer’s suggestion, we have incorporated fine-tuned versions of scREBOUND, scGPT, and scFoundation into our evaluation.
>
> We evaluated the fine-tuned models on three major benchmarking tasks: (1) batch effect removal, (2) cell type annotation, and (3) data imputation. The corresponding results are presented in Tables 7~9. scREBOUND still shows superior performance in the fine-tuned setting.
>
> Table 7. The evaluation of batch effect removal on fine-tuned models
> |Methods|Immune|||||Pancreas|||||Lung|||||Covid19|||||CL|||||PBMC 1|||||Pancrm|||||PBMC 2|||||
> |-|-|-|-|-|-|-|-|-|-|-|-|-|-|-|-|-|-|-|-|-|-|-|-|-|-|-|-|-|-|-|-|-|-|-|-|-|-|-|-|-|
> ||ARI|NMI|ASW|ASW-Batch|GC|ARI|NMI|ASW|ASW-Batch|GC|ARI|NMI|ASW|ASW-Batch|GC|ARI|NMI|ASW|ASW-Batch|GC|ARI|NMI|ASW|ASW-Batch|GC|ARI|NMI|ASW|ASW-Batch|GC|ARI|NMI|ASW|ASW-Batch|GC|ARI|NMI|ASW|ASW-Batch|GC|
> |scREBOUND-finetune|0.61|0.76|0.61|0.77|0.70|0.66|0.69|0.61|0.81|0.82|0.56|0.70|0.58|0.79|0.67|0.63|0.69|0.66|0.70|0.49|0.69|0.71|0.88|0.74|1.00|0.41|0.64|0.64|0.69|0.86|0.43|0.57|0.54|0.74|0.88|0.74|0.72|0.60|0.84|0.88|
> |scGPT-finetune|0.54|0.68|0.53|0.83|0.88|0.26|0.44|0.47|0.82|0.89|0.51|0.66|0.56|0.80|0.75|0.63|0.63|0.58|0.73|0.63|0.41|0.43|0.58|0.87|1.00|0.32|0.41|0.46|0.85|1.00|0.20|0.45|0.35|0.57|0.88|0.26|0.34|0.51|0.82|1.00|
> |scFoundation-finetune|0.56|0.74|0.61|0.79|0.81|0.53|0.65|0.56|0.84|0.93|0.49|0.71|0.57|0.80|0.79|0.37|0.60|0.59|0.76|0.69|0.76|0.66|0.58|0.91|1.00|0.29|0.42|0.47|0.79|1.00|0.20|0.46|0.38|0.56|0.92|0.10|0.22|0.49|0.89|0.99|
>
> Table 8. The evaluation of cell type annotation on fine-tuned models
> |Methods|Immune|||Pancreas|||Lung|||Covid19|||PBMC1|||PancRM|||PBMC|||
> |-|-|-|-|-|-|-|-|-|-|-|-|-|-|-|-|-|-|-|-|-|-|
> ||Acc|F1(Unw)|F1(W)|Acc|F1(Unw)|F1(W)|Acc|F1(Unw)|F1(W)|Acc|F1(Unw)|F1(W)|Acc|F1(Unw)|F1(W)|Acc|F1(Unw)|F1(W)|Acc|F1(Unw)|F1(W)|
> |scREBOUND-finetune|0.81|0.77|0.79|0.84|0.63|0.80|0.78|0.72|0.78|0.91|0.91|0.91|0.92|0.82|0.92|0.81|0.61|0.75|0.92|0.85|0.92|
> |scFoundation-finetune|0.88|0.82|0.85|0.87|0.72|0.82|0.92|0.89|0.92|0.95|0.95|0.95|0.98|0.95|0.98|0.91|0.80|0.89|0.94|0.90|0.94|
> |scGPT-finetune|0.78|0.64|0.75|0.49|0.23|0.43|0.68|0.67|0.63|0.92|0.92|0.93|0.47|0.33|0.37|0.19 |0.04|0.11|0.26|0.12|0.21|
> |scMulan-finetune|0.86|0.80|0.83|0.85|0.71|0.81|0.91|0.88|0.92|0.95|0.95|0.95|0.97|0.93|0.97|0.89|0.76|0.86|0.92|0.85|0.92|
>
> Table 9. The evaluation of data imputation on fine-tuned models
> |Mask|Method|Immune||Pancreas||Lung||Covid19||CL||HumanPBMC||PancRM||PBMC_Raw||
> |-|-|-|-|-|-|-|-|-|-|-|-|-|-|-|-|-|-|
> |||Pcorr|Scorr|Pcorr|Scorr|Pcorr|Scorr|Pcorr|Scorr|Pcorr|Scorr|Pcorr|Scorr|Pcorr|Scorr|Pcorr|Scorr|
> |10%|scREBOUND-finetune|0.9314|0.3841|0.6250|0.4986|0.7298|0.3950|0.8223|0.3734|0.9661|0.5585|0.9414|0.3863|0.6615|0.5302|0.8852|0.2736|
> ||scGPT-finetune|-0.0002|-0.0002|0.0001|0.0002|-0.0002|-0.0007|0.0003|-0.0003|0.0010|0.0008|-0.0001|0.0004|0.0005|0.0008|-0.0004|-0.0011|
> |20%|scREBOUND-finetune|0.9300|0.3860|0.6274|0.4979|0.7484|0.4002|0.8367|0.3739|0.9679|0.5603|0.9441|0.3897|0.6727|0.5268|0.8943|0.2739|
> ||scGPT-finetune|-0.0003|-0.0003|0.0004|-0.0001|-0.0002|-0.0007|0.0000|-0.0004|0.0005|0.0004|0.0002|0.0004|0.0005|0.0008|-0.0004|-0.0010|

---

### Official Review · Reviewer_UZvz · 2025-11-01

**Soundness:** 3
**Presentation:** 3
**Contribution:** 2
**Rating:** 2
**Confidence:** 4

**Summary:**

This paper presents scREBOUND, a computationally efficient single-cell foundation model designed to address batch effects across experiments through a batch embedding network and multi-granular contrastive loss regularization. The authors introduce a protein-language-model-informed gene compression strategy that reduces gene tokens per cell while preserving biological information. scREBOUND was evaluated on four downstream tasks.

**Strengths:**

- The paper demonstrates substantial improvements in computational performance, achieving up to 17x reduction in inference time and 30x reduction in memory usage compared to state-of-the-art models like scGPT.
- The evaluation is, for the most part, well done and comprehensive. It includes most of the state-of-the-art foundation models and evaluates them on four zero-shot tasks across eight different datasets. Furthermore, the paper includes several ablations to disentangle the performance gains attributed to different complements of the model.
- Using summary statistics of batches to embed them is an interesting idea.

**Weaknesses:**

- Compressing genes to meta genes using a pLM was already introduced in [1], which isn't cited in the paper. This severely limited the paper's novelty. Furthermore, the authors should compare their approach to SATURN, especially for the cross-species transfer task. As the pLM informed meta gene module allows for the mapping of (protein-coding) genes across species to meta-genes directly.
- The batch correction evaluation is only done on ARI, NMI, and ASW. It only measures how well the biological signal is retained in the latent representation. However, [2] introduces two axes for evaluating batch correction, batch integration, and biological conservation. In the paper, the batch integration metrics are missing. Figures 6 and 7 clearly demonstrate that batch effects persist in scREBOUND's latent representation.
- [3] shows that reducing the size of scFMs can help improve performance on several downstream tasks.
- The results tables present point estimates without standard errors or confidence intervals, making it difficult to assess the statistical significance of performance differences between scREBOUND and baseline methods. For example, computing ARI and NMI using Leiden clustering can yield very different values depending on the random seed.
- All evaluations in the paper are done in a zero-shot fashion. Several tasks benefit significantly from fine-tuning. Therefore, it would be beneficial to include fine-tuning of the models in the evaluation.
- It is unclear how the features for the batch encoder were chosen. An ablation on them would be very interesting. Furthermore, the embedding of the batch encoder could be explored more in the paper to understand what kind of representations the model learn for different batches.
- Figures 4 and 5 suggest that most of the performance gains come from the addition of the contrastive loss rather than the batch encoder. The contrastive loss requires cell type labels during pretraining, whereas the other models did not use them. This could lead to issues for underrepresented cell types in the training data.
- Theorem 1 feels disconnected from the rest of the paper and the empirical results.

[1] Rosen, Yanay, et al. "Toward universal cell embeddings: integrating single-cell RNA-seq datasets across species with SATURN." Nature Methods 21.8 (2024): 1492-1500.
[2] Luecken, Malte D., et al. "Benchmarking atlas-level data integration in single-cell genomics." Nature methods 19.1 (2022): 41-50.
[3] Theus, Alexander, et al. "CancerFoundation: A single-cell RNA sequencing foundation model to decipher drug resistance in cancer." bioRxiv (2024): 2024-11.

**Questions:**

- How does scREBOUND compare to SATURN. How different is the meta-gene module proposed in SATURN from scREBOUND's?
- How does scREBOUND perform on batch integration metrics (kBET, AWS (batch-label), ...) compared to other methods?
- How does Theorem 1 relate to the empirical findings of the paper and compressing genes using a pML?
- How were the features for the batch encoder selected? How does the performance change when we change the set of selected features? (What kind of representation of batches did the batch encoder learn?)
- How was the cell type overlap between the 1 million randomly selected cells and the cells in the test set for zero-shot cell type annotation? Shouldn't several cell types not be represented in the randomly selected training set?
- scGPTs' performance seems to improve with a higher percentage of masked genes for imputation (Table 6). Is there a potential reason for this?

---

> ### Author Response · Authors · 2025-12-03
>
> ```
> Q1. Compressing genes to meta genes using a pLM was already introduced in [1], which isn't cited in the paper. This severely limited the paper's novelty. Furthermore, the authors should compare their approach to SATURN, especially for the cross-species transfer task. As the pLM informed meta gene module allows for the mapping of (protein-coding) genes across species to meta-genes directly. How does scREBOUND compare to SATURN. How different is the meta-gene module proposed in SATURN from scREBOUND's?
> ```
> **A1**. We agree that SATURN also uses protein language model, but it serves completely different purposes.
>
> SATURN uses protein embeddings only to align orthologous genes across species within an autoencoding architecture. It does not perform gene compression: the full gene space is retained, and the alignment is re-learned in an unsupervised manner for each new dataset.
>
> scREBOUND uses protein embeddings to construct a stable, reusable compression module that groups functionally similar genes into meta-genes and substantially reduces computational cost. The embeddings are integrated into a transformer-based compression mechanism, and the resulting compressed representation can be applied directly to diverse downstream datasets without retraining the alignment.
>
> According to the reviewers suggestion, we compared the performance of SATURN with scREBOUND. Since SATURN is designed for multi-species integration, we compare the performance of scREBOUND and SATURN on the task of cross-species knowledge transfer (result in Table 1). Since SATURN is an unsupervised model, we train SATURN with human and mouse data, and then evaluate its performance on held-out mouse dataset. The evaluation shows that SATURN does not generalize well to the test dataset, and scREBOUND has a significantly better performance compared to SATURN.
>
> Table 1: Scores of different model on mouse data knowledge transfer
> |Methods|Mouse data||||||
> |-|-|-|-|-|-|-|
> ||ARI|NMI|ASW|ASW (batch)|Graph Connectivity|
> |scREBOUND|0.53|0.65|0.55|0.59|0.77|
> |scGPT|0.27|0.44|0.46|0.60|0.50|
> |scMulan|0.35|0.57|0.54|0.61|0.62|
> |UCE|0.41|0.67|0.56|0.50|0.85|
> |scFoundation|0.52|0.66|0.59|0.44|0.83|
> |SATURN|0.22|0.35|0.50|0.76|0.53|

---

> ### Author Response · Authors · 2025-12-03
>
> ```
> Q2. The batch correction evaluation is only done on ARI, NMI, and ASW. It only measures how well the biological signal is retained in the latent representation. However, [2] introduces two axes for evaluating batch correction, batch integration, and biological conservation. In the paper, the batch integration metrics are missing. Figures 6 and 7 clearly demonstrate that batch effects persist in scREBOUND's latent representation. How does scREBOUND perform on batch integration metrics (kBET, AWS (batch-label), ...) compared to other methods?
> ```
> **A2**. We thank the reviewer for the suggestion. As the reviewer mentioned, the benchmarking paper consider two axes of metrics for biological conservation and batch correction, and the three metrics we used in the paper are from the biological conservation side. According to the reviewer's suggestion, we incorporate two additional metrics from the batch correction side, including ASW (batch) and Graph Connectivity Score. We updated the score in the manuscript and post it in Table 2 for the reviewer's reference.
>
>
> Table 2: Updated scores on zero-shot batch effect removal with error bar and new batch integration metrics
> |Methods|Immune|||||Pancreas|||||Lung|||||Covid19|||||CL|||||HumanPBMC|||||PancRM|||||PBMC_Raw|||||
> |-|-|-|-|-|-|-|-|-|-|-|-|-|-|-|-|-|-|-|-|-|-|-|-|-|-|-|-|-|-|-|-|-|-|-|-|-|-|-|-|-|
> ||ARI|NMI|ASW|ASW-Batch|GC|ARI|NMI|ASW|ASW-Batch|GC|ARI|NMI|ASW|ASW-Batch|GC|ARI|NMI|ASW|ASW-Batch|GC|ARI|NMI|ASW|ASW-Batch|GC|ARI|NMI|ASW|ASW-Batch|GC|ARI|NMI|ASW|ASW-Batch|GC|ARI|NMI|ASW|ASW-Batch|GC|
> |scREBOUND|0.54 ± 0.052|0.74 ± 0.009|0.61|0.85|0.97|0.60 ± 0.001|0.74 ± 0.001|0.60|0.86|0.89|0.51 ± 0.034|0.70 ± 0.002|0.58|0.88|0.97|0.46 ± 0.004|0.63 ± 0.003|0.64|0.85|0.97|0.96 ± 0.000|0.91 ± 0.000|0.80|0.89|0.99|0.57 ± 0.012|0.70 ± 0.005|0.65|0.84|0.90|0.65 ± 0.001|0.75 ± 0.001|0.60|0.84|0.90|0.80 ± 0.004|0.79 ± 0.002|0.68|0.83|0.95|
> |scGPT|0.48 ± 0.009|0.68 ± 0.003|0.56|0.87|0.97|0.35 ± 0.027|0.52 ± 0.014|0.52|0.88|0.92|0.49 ± 0.010|0.73 ± 0.003|0.57|0.88|0.97|0.63 ± 0.024|0.64 ± 0.014|0.56|0.87|0.97|0.87 ± 0.033|0.79 ± 0.042|0.58|0.93|1.00|0.54 ± 0.010|0.63 ± 0.011|0.51|0.86|0.86|0.34 ± 0.004|0.43 ± 0.003|0.49|0.78|0.80|0.37 ± 0.052|0.51 ± 0.024|0.51|0.80|0.93|
> |scMulan|0.35 ± 0.015|0.59 ± 0.005|0.55|0.88|0.67|0.22 ± 0.014|0.42 ± 0.003|0.48|0.81|0.67|0.45 ± 0.045|0.67 ± 0.004|0.55|0.86|0.73|0.30 ± 0.020|0.53 ± 0.005|0.58|0.87|0.69|0.32 ± 0.028|0.46 ± 0.010|0.65|0.97|0.72|0.32 ± 0.029|0.57 ± 0.003|0.58|0.75|0.66|0.33 ± 0.044|0.60 ± 0.002|0.51|0.84|0.74|0.37 ± 0.002|0.62 ± 0.001|0.59|0.87|0.93|
> |UCE|0.33 ± 0.050|0.64 ± 0.005|0.55|0.88|0.98|0.43 ± 0.026|0.62 ± 0.011|0.55|0.89|0.95|0.43 ± 0.026|0.67 ± 0.003|0.55|0.92|0.96|0.29 ± 0.009|0.57 ± 0.001|0.57|0.90|0.90|0.82 ± 0.001|0.79 ± 0.001|0.74|0.95|0.99|0.44 ± 0.042|0.68 ± 0.010|0.58|0.80|0.98|0.50 ± 0.003|0.62 ± 0.001|0.56|0.82|0.91|0.82 ± 0.016|0.78 ± 0.011|0.55|0.88|0.98|
> |scFoundation|0.36 ± 0.031|0.66 ± 0.005|0.57|0.81|0.88|0.37 ± 0.004|0.57 ± 0.005|0.53|0.82|0.90|0.39 ± 0.007|0.72 ± 0.004|0.55|0.88|0.89|0.27 ± 0.039|0.54 ± 0.007|0.56|0.90|0.87|0.71 ± 0.000|0.74 ± 0.000|0.79|0.81|0.99|0.42 ± 0.026|0.68 ± 0.010|0.58|0.68|0.86|0.30 ± 0.012|0.59 ± 0.002|0.55|0.78|0.89|0.55 ± 0.007|0.71 ± 0.009|0.59|0.80|0.98|

---

> ### Author Response · Authors · 2025-12-03
>
> ```
> Q3. The results tables present point estimates without standard errors or confidence intervals, making it difficult to assess the statistical significance of performance differences between scREBOUND and baseline methods. For example, computing ARI and NMI using Leiden clustering can yield very different values depending on the random seed.
> ```
> **A3**. According to the reviewers suggestion, we added the standard errors and average score across multiple random seeds into our evaluation test. The updated result of batch effect removal test is shown in Table 2. We run Leiden clustering with 5 random seeds, and calculate the mean and standard deviation of ARI, NMI, ASW, ASW-Batch, and GC scores. We also updates the scores of tasks including cell type annotation, data imputation, cross-species knowledge transfer to include the standard error (Tables 1, 3, 4). The scores without the standard errors are deterministic and are the same across runs.
>
> **Table 3**. Updated scores on zero-shot cell type annotation with error bar
> |Classifier|Method|Immune||Pancreas||Lung||Covid19||Humanpbmc||PancRM||PBMC_Raw||
> |-|-|-|-|-|-|-|-|-|-|-|-|-|-|-|-|
> |||WF1|Acc|WF1|Acc|WF1|Acc|WF1|Acc|WF1|Acc|WF1|Acc|WF1|Acc|
> |knn_5|scREBOUND|0.723±0.017|0.703±0.019|0.800±0.013|0.813±0.008|0.512±0.011|0.582±0.007|0.699±0.021|0.714±0.015|0.706±0.010|0.707±0.008|0.714±0.024|0.735±0.015|0.480±0.020|0.481±0.013|
> ||scFoundation|0.695±0.012|0.696±0.014|0.700±0.009|0.738±0.005|0.548±0.007|0.621±0.007|0.681±0.007|0.687±0.006|0.687±0.010|0.685±0.008|0.636±0.023|0.671±0.021|0.460±0.057|0.478±0.037|
> ||scGPT|0.285±0.066|0.328±0.063|0.307±0.024|0.317±0.036|0.286±0.055|0.292±0.047|0.381±0.022|0.422±0.035|0.578±0.020|0.566±0.015|0.328±0.020|0.313±0.025|0.312±0.078|0.338±0.056|
> ||scMulan|0.665±0.027|0.682±0.039|0.081±0.012|0.111±0.014|0.568±0.002|0.638±0.003|0.686±0.009|0.707±0.008|0.689±0.019|0.696±0.017|0.672±0.012|0.698±0.012|0.470±0.024|0.475±0.017|
> |svm|scREBOUND|0.779±0.007|0.783±0.007|0.883±0.015|0.881±0.014|0.525±0.013|0.549±0.020|0.773±0.009|0.778±0.008|0.807±0.009|0.810±0.007|0.841±0.024|0.841±0.022|0.593±0.021|0.592±0.015|
> ||scFoundation|0.691±0.036|0.697±0.054|0.798±0.014|0.827±0.012|0.545±0.005|0.617±0.004|0.764±0.010|0.770±0.009|0.779±0.026|0.776±0.022|0.742±0.027|0.780±0.017|0.453±0.058|0.482±0.037|
> ||scGPT|0.038±0.051|0.099±0.130|0.000±0.000|0.011±0.008|0.079±0.043|0.204±0.096|0.056±0.070|0.147±0.134|0.582±0.017|0.638±0.016|0.505±0.036|0.517±0.045|0.082±0.023|0.156±0.048|
> ||scMulan|0.677±0.016|0.677±0.021|0.173±0.044|0.227±0.031|0.595±0.006|0.637±0.006|0.823±0.009|0.827±0.008|0.847±0.012|0.847±0.012|0.807±0.008|0.820±0.007|0.572±0.010|0.577±0.008|
>
> **Table 4**. Updated scores on zero-shot imputation with error bar
>
> |Mask|Method|Immune||Pancreas||Lung||Covid19||CL||HumanPBMC||PancRM||PBMC_Raw||
> |-|-|-|-|-|-|-|-|-|-|-|-|-|-|-|-|-|-|
> |||Pcorr|Scorr|Pcorr|Scorr|Pcorr|Scorr|Pcorr|Scorr|Pcorr|Scorr|Pcorr|Scorr|Pcorr|Scorr|Pcorr|Scorr|
> |10%|scREBOUND|0.8988±0.0026|0.3705±0.0016|0.4333±0.0039|0.4307±0.0027|0.6055±0.0050|0.3710±0.0023|0.7838±0.0021|0.3672±0.0011|0.8351±0.0020|0.5096±0.0013|0.9204±0.0021|0.3772±0.0017|0.4504±0.0063|0.4539±0.0019|0.8255±0.0032|0.2666±0.0008|
> ||scGPT|0.0003±0.0004|0.0004±0.0003|-0.0004±0.0004|0.0003±0.0005|0.0000±0.0003|0.0002±0.0003|0.0009±0.0004|0.0006±0.0006|0.0008±0.0004|0.0002±0.0006|0.0000±0.0005|0.0002±0.0003|-0.0006±0.0007|0.0000±0.0004|0.0003±0.0002|0.0001±0.0005|
> ||scFoundation|0.8498±0.0020|0.3606±0.0009|0.5992±0.0029|0.4149±0.0022|0.6859±0.0043|0.3794±0.0022|0.7527±0.0025|0.3618±0.0009|0.7786±0.0007|0.4864±0.0006|0.8604±0.0007|0.3723±0.0015|0.6150±0.0028|0.4448±0.0010|0.8342±0.0027|0.2669±0.0008|
> |20%|scREBOUND|0.8883±0.0033|0.3699±0.0013|0.3720±0.0047|0.4271±0.0039|0.5941±0.0054|0.3710±0.0016|0.7806±0.0031|0.3666±0.0007|0.8277±0.0009|0.5109±0.0008|0.9178±0.0027|0.3776±0.0014|0.3989±0.0061|0.4493±0.0022|0.8249±0.0028|0.2669±0.0007|
> ||scGPT|0.0002±0.0001|0.0003±0.0004|-0.0001±0.0005|0.0000±0.0003|-0.0001±0.0004|0.0000±0.0005|0.0006±0.0004|0.0004±0.0005|0.0002±0.0005|-0.0003±0.0002|0.0003±0.0001|0.0002±0.0004|-0.0004±0.0004|-0.0000±0.0003|0.0004±0.0003|0.0002±0.0004|
> ||scFoundation|0.8395±0.0034|0.3604±0.0008|0.5354±0.0039|0.4164±0.0018|0.6816±0.0040|0.3803±0.0019|0.7398±0.0031|0.3634±0.0007|0.7770±0.0006|0.4820±0.0008|0.8534±0.0009|0.3740±0.0012|0.5554±0.0048|0.4472±0.0014|0.8244±0.0019|0.2672±0.0007|

---

> ### Author Response · Authors · 2025-12-03
>
> ```
> Q4. All evaluations in the paper are done in a zero-shot fashion, it would be beneficial to include fine-tuning of the models in the evaluation.
> ```
> **A4**.  Following the reviewer’s suggestion, we have incorporated fine-tuned versions of scREBOUND and baseline models into our evaluation. We evaluated the fine-tuned models on (1) batch effect removal, (2) cell type annotation, and (3) data imputation. The corresponding results are presented in Tables 5~7. scREBOUND still shows superior performance in the fine-tuned setting.
>
> **Table 5**. The evaluation of batch effect removal on fine-tuned models
> |Methods|Immune|||||Pancreas|||||Lung|||||Covid19|||||CL|||||HumanPBMC|||||PancRM|||||PBMC_Raw|||||
> |-|-|-|-|-|-|-|-|-|-|-|-|-|-|-|-|-|-|-|-|-|-|-|-|-|-|-|-|-|-|-|-|-|-|-|-|-|-|-|-|-|
> ||ARI|NMI|ASW|ASW-B|GC|ARI|NMI|ASW|ASW-B|GC|ARI|NMI|ASW|ASW-B|GC|ARI|NMI|ASW|ASW-B|GC|ARI|NMI|ASW|ASW-B|GC|ARI|NMI|ASW|ASW-B|GC|ARI|NMI|ASW|ASW-B|GC|ARI|NMI|ASW|ASW-B|GC|
> |scREBOUND-finetune|0.61±0.01|0.76±0.00|0.61±0.00|0.77±0.00|0.70±0.00|0.66±0.03|0.69±0.01|0.61±0.00|0.81±0.00|0.82±0.00|0.56±0.00|0.70±0.00|0.58±0.00|0.79±0.00|0.67±0.00|0.63±0.01|0.69±0.01|0.66±0.00|0.70±0.00|0.49±0.00|0.69±0.02|0.71±0.02|0.88±0.00|0.74±0.00|1.00±0.00|0.41±0.00|0.64±0.00|0.64±0.00|0.69±0.00|0.86±0.00|0.43±0.04|0.57±0.01|0.54±0.00|0.74±0.00|0.88±0.00|0.74±0.00|0.72±0.00|0.60±0.00|0.84±0.00|0.88±0.00|
> |scGPT-finetune|0.54±0.00|0.68±0.00|0.53±0.00|0.83±0.00|0.88±0.00|0.26±0.02|0.44±0.01|0.47±0.00|0.82±0.00|0.89±0.00|0.51±0.01|0.66±0.00|0.56±0.00|0.80±0.00|0.75±0.00|0.63±0.01|0.63±0.01|0.58±0.00|0.73±0.00|0.63±0.00|0.41±0.02|0.43±0.01|0.58±0.00|0.87±0.00|1.00±0.00|0.32±0.02|0.41±0.01|0.46±0.00|0.85±0.00|1.00±0.00|0.20±0.02|0.45±0.01|0.35±0.00|0.57±0.00|0.88±0.00|0.26±0.01|0.34±0.01|0.51±0.00|0.82±0.00|1.00±0.00|
> |scMulan-finetune|0.48±0.00|0.60±0.00|0.54±0.00|0.78±0.00|0.50±0.00|0.17±0.01|0.37±0.00|0.37±0.00|0.71±0.00|0.56±0.00|0.49±0.02|0.64±0.00|0.55±0.00|0.79±0.00|0.62±0.00|0.38±0.01|0.57±0.00|0.60±0.00|0.73±0.00|0.30±0.00|0.19±0.04|0.22±0.03|0.55±0.00|0.90±0.00|1.00±0.00|0.27±0.02|0.45±0.01|0.47±0.00|0.75±0.00|1.00±0.00|0.21±0.01|0.48±0.00|0.42±0.00|0.55±0.00|0.92±0.00|0.22±0.01|0.27±0.01|0.49±0.00|0.87±0.00|0.99±0.00|
> |UCE-finetune|0.24±0.03|0.54±0.01|0.46±0.00|0.69±0.00|0.83±0.00|0.35±0.03|0.51±0.01|0.50±0.00|0.82±0.00|0.90±0.00|0.33±0.01|0.53±0.01|0.45±0.00|0.65±0.00|0.83±0.00|0.43±0.01|0.62±0.01|0.63±0.00|0.74±0.00|0.67±0.00|0.72±0.01|0.62±0.00|0.59±0.00|0.88±0.00|1.00±0.00|0.39±0.06|0.49±0.02|0.48±0.00|0.81±0.00|1.00±0.00|0.20±0.01|0.51±0.00|0.44±0.00|0.57±0.00|0.92±0.00|0.24±0.06|0.29±0.01|0.51±0.00|0.84±0.00|0.99±0.00|
> |scFoundation-finetune|0.56±0.00|0.74±0.00|0.61±0.00|0.79±0.00|0.81±0.00|0.53±0.04|0.72±0.00|0.56±0.00|0.84±0.00|0.93±0.00|0.49±0.01|0.66±0.01|0.57±0.00|0.80±0.00|0.79±0.00|0.37±0.02|0.60±0.00|0.59±0.00|0.76±0.00|0.69±0.00|0.76±0.01|0.66±0.01|0.58±0.00|0.91±0.00|1.00±0.00|0.29±0.02|0.54±0.01|0.47±0.00|0.79±0.00|1.00±0.00|0.20±0.02|0.58±0.01|0.38±0.00|0.56±0.00|0.92±0.00|0.10±0.06|0.22±0.00|0.49±0.00|0.89±0.00|0.99±0.00|
>
> **Table 6**. The evaluation of cell type annotation on fine-tuned models
>
> |Methods|Immune||Pancreas||Lung||Covid19||HumanPBMC||PancRM||PBMC||
> |-|-|-|-|-|-|-|-|-|-|-|-|-|-|-|
> ||WF1|Acc|WF1|Acc|WF1|Acc|WF1|Acc|WF1|Acc|WF1|Acc|WF1|Acc|
> |scREBOUND-finetune|0.792 ± 0.032|0.811 ± 0.038|0.797 ± 0.003|0.845 ± 0.003|0.782 ± 0.059|0.778 ± 0.047|0.912 ± 0.002|0.913 ± 0.002|0.921 ± 0.007|0.923 ± 0.007|0.752 ± 0.008|0.812 ± 0.007|0.922 ± 0.006|0.918 ± 0.006|
> |scFoundation-finetune|0.848 ± 0.005|0.875 ± 0.006|0.819 ± 0.001|0.867 ± 0.001|0.923 ± 0.004|0.921 ± 0.005|0.951 ± 0.002|0.951 ± 0.002|0.979 ± 0.001|0.978 ± 0.001|0.888 ± 0.005|0.913 ± 0.006|0.945 ± 0.009|0.944 ± 0.008|
> |scGPT-finetune|0.752 ± 0.096|0.784 ± 0.084|0.428 ± 0.080|0.487 ± 0.073|0.628 ± 0.001|0.684 ± 0.004|0.925 ± 0.002|0.924 ± 0.002|0.371 ± 0.021|0.469 ± 0.006|0.112 ± 0.088|0.194 ± 0.138|0.214 ± 0.204|0.257 ± 0.218|
> |scMulan-finetune|0.835 ± 0.006|0.861 ± 0.009|0.812 ± 0.040|0.847 ± 0.024|0.917 ± 0.002|0.914 ± 0.002|0.948 ± 0.004|0.948 ± 0.004|0.969 ± 0.001|0.969 ± 0.001|0.860 ± 0.006|0.887 ± 0.005|0.919 ± 0.011|0.916 ± 0.011|

---

> > ### Author Response · Authors · 2025-12-03
> >
> > **Table 7**. The evaluation of data imputation on fine-tuned models
> >
> > |Mask|Method|Immune||Pancreas||Lung||Covid19||CL||HumanPBMC||PancRM||PBMC_Raw||
> > |-|-|-|-|-|-|-|-|-|-|-|-|-|-|-|-|-|-|
> > |||Pcorr|Scorr|Pcorr|Scorr|Pcorr|Scorr|Pcorr|Scorr|Pcorr|Scorr|Pcorr|Scorr|Pcorr|Scorr|Pcorr|Scorr|
> > |10%|scREBOUND-finetune|0.9314 ± 0.0085|0.3841 ± 0.0063|0.6250 ± 0.0850|0.4986 ± 0.0145|0.7298 ± 0.0166|0.3950 ± 0.0055|0.8223 ± 0.0322|0.3734 ± 0.0044|0.9661 ± 0.0058|0.5585 ± 0.0056|0.9414 ± 0.0018|0.3863 ± 0.0032|0.6615 ± 0.0498|0.5302 ± 0.0017|0.8852 ± 0.0034|0.2736 ± 0.0004|
> > ||scGPT-finetune|-0.0002 ± 0.0004|-0.0002 ± 0.0003|0.0001 ± 0.0004|0.0002 ± 0.0005|-0.0002 ± 0.0003|-0.0007 ± 0.0003|0.0003 ± 0.0004|-0.0003 ± 0.0006|0.0010 ± 0.0004|0.0008 ± 0.0006|-0.0001 ± 0.0005|0.0004 ± 0.0003|0.0005 ± 0.0007|0.0008 ± 0.0004|-0.0004 ± 0.0002|-0.0011 ± 0.0005|
> > |20%|scREBOUND-finetune|0.9300 ± 0.0036|0.3860 ± 0.0051|0.6274 ± 0.0782|0.4979 ± 0.0060|0.7484 ± 0.0088|0.4002 ± 0.0057|0.8367 ± 0.0145|0.3739 ± 0.0051|0.9679 ± 0.0020|0.5603 ± 0.0023|0.9441 ± 0.0026|0.3897 ± 0.0033|0.6727 ± 0.0338|0.5268 ± 0.0086|0.8943 ± 0.0011|0.2739 ± 0.0021|
> > ||scGPT-finetune|-0.0003 ± 0.0001|-0.0003 ± 0.0004|0.0004 ± 0.0005|-0.0001 ± 0.0003|-0.0002 ± 0.0004|-0.0007 ± 0.0005|0.0000 ± 0.0004|-0.0004 ± 0.0004|0.0005 ± 0.0006|0.0004 ± 0.0002|0.0002 ± 0.0001|0.0004 ± 0.0004|0.0005 ± 0.0003|0.0008 ± 0.0003|-0.0004 ± 0.0003|-0.0010 ± 0.0004|

---

> ### Author Response · Authors · 2025-12-03
>
> ```
> Q5. It is unclear how the features for the batch encoder were chosen. An ablation on them would be very interesting. Furthermore, the embedding of the batch encoder could be explored more in the paper to understand what kind of representations the model learn for different batches. How were the features for the batch encoder selected? How does the performance change when we change the set of selected features? (What kind of representation of batches did the batch encoder learn?)
> ```
> **A5**. We hand pick the batch-related features that were reported to highly correlate with the batch effect of the datasets. The features we considered includes 4 categories: 1. batch-specific expression distribution statistics, 2. expression level of house keeping genes, 3. expression level of ribosomal genes, and 4. expression level of stress-related genes. The distribution statistics considers average proportion of non-zero genes within each batch, average expression of non-zero genes within each batch, and average proportion of mito-genes within each batch. These statistics are all heavily affected by the batch effect. In addition, the house keeping genes, ribosomal genes, and stress-related genes were all reported to be highly-correlated with batch effect (see the supplementary material A5 for details).
>
> We agree that an ablation test on the batch features is important to justify the selection of these features for the batch encoding network. We train scREBOUND again with batch encoder that only consider each one of the four feature categories. We evaluate the model performance using the task of batch effect removal, since the batch encoding is used to remove the batch effect. The result (Table 8) shows that scREBOUND-original provides more stable and superior performance compared to the baseline methods in removing batch effect.
>
> Table 8. Batch effect removal
> |Methods|Immune|||||Pancreas|||||Lung|||||Covid19|||||CL|||||HumanPBMC|||||PancRM|||||PBMC_Raw|||||
> |-|-|-|-|-|-|-|-|-|-|-|-|-|-|-|-|-|-|-|-|-|-|-|-|-|-|-|-|-|-|-|-|-|-|-|-|-|-|-|-|-|
> ||ARI|NMI|ASW|ASW-Batch|GC|ARI|NMI|ASW|ASW-Batch|GC|ARI|NMI|ASW|ASW-Batch|GC|ARI|NMI|ASW|ASW-Batch|GC|ARI|NMI|ASW|ASW-Batch|GC|ARI|NMI|ASW|ASW-Batch|GC|ARI|NMI|ASW|ASW-Batch|GC|ARI|NMI|ASW|ASW-Batch|GC|
> |scREBOUND-original|0.55 ± 0.05|0.74 ± 0.01|0.61|0.85|0.84|0.61 ± 0.02|0.74 ± 0.01|0.60|0.86|0.94|0.51 ± 0.02|0.70 ± 0.00|0.58|0.88|0.81|0.46 ± 0.00|0.62 ± 0.00|0.64|0.85|0.60|0.96 ± 0.00|0.91 ± 0.00|0.80|0.89|1.00|0.57 ± 0.02|0.70 ± 0.01|0.65|0.84|1.00|0.65 ± 0.00|0.75 ± 0.00|0.60|0.84|0.98|0.80 ± 0.00|0.79 ± 0.00|0.68|0.83|1.00|
> |scREBOUND-Bstats|0.62 ± 0.02|0.76 ± 0.01|0.61|0.86|0.97|0.47 ± 0.00|0.67 ± 0.00|0.57|0.87|0.91|0.53 ± 0.01|0.71 ± 0.00|0.58|0.88|0.97|0.44 ± 0.00|0.61 ± 0.00|0.64|0.85|0.97|0.86 ± 0.14|0.84 ± 0.10|0.78|0.90|0.99|0.45 ± 0.00|0.67 ± 0.00|0.65|0.83|0.89|0.60 ± 0.01|0.71 ± 0.01|0.56|0.84|0.89|0.81 ± 0.00|0.78 ± 0.00|0.68|0.82|0.95|
> |scREBOUND-BHK|0.62 ± 0.00|0.78 ± 0.00|0.61|0.87|0.96|0.59 ± 0.04|0.73 ± 0.02|0.57|0.86|0.90|0.50 ± 0.02|0.70 ± 0.00|0.57|0.88|0.97|0.45 ± 0.00|0.62 ± 0.00|0.64|0.85|0.97|0.97 ± 0.00|0.94 ± 0.00|0.79|0.90|0.99|0.57 ± 0.01|0.69 ± 0.01|0.65|0.85|0.90|0.61 ± 0.02|0.73 ± 0.01|0.57|0.85|0.92|0.63 ± 0.15|0.73 ± 0.03|0.68|0.82|0.96|
> |scREBOUND-BRibo|0.60 ± 0.02|0.76 ± 0.01|0.61|0.86|0.96|0.51 ± 0.03|0.69 ± 0.01|0.58|0.85|0.89|0.52 ± 0.00|0.71 ± 0.01|0.58|0.88|0.97|0.46 ± 0.00|0.62 ± 0.00|0.64|0.84|0.97|0.96 ± 0.00|0.93 ± 0.01|0.80|0.88|0.99|0.57 ± 0.00|0.69 ± 0.00|0.65|0.85|0.90|0.61 ± 0.00|0.72 ± 0.00|0.58|0.84|0.90|0.56 ± 0.12|0.73 ± 0.03|0.69|0.81|0.95|
> |scREBOUND-BStress|0.60 ± 0.03|0.75 ± 0.01|0.61|0.86|0.96|0.46 ± 0.01|0.67 ± 0.00|0.56|0.87|0.90|0.51 ± 0.02|0.71 ± 0.00|0.58|0.88|0.97|0.45 ± 0.01|0.61 ± 0.01|0.64|0.85|0.97|0.96 ± 0.00|0.91 ± 0.00|0.79|0.89|1.00|0.44 ± 0.03|0.67 ± 0.01|0.65|0.83|0.90|0.55 ± 0.01|0.69 ± 0.01|0.57|0.85|0.91|0.80 ± 0.01|0.78 ± 0.01|0.68|0.82|0.95|
>
> ```
> Q6. The contrastive loss requires cell type labels during pretraining, whereas the other models did not use them. This could lead to issues for underrepresented cell types in the training data.
> ```
> **A6**. We thank the reviewer for raising the question. First, we would like to clarify that several of the baseline models that we compare also use the cell type labels in the pre-training: scGPT model is continual pretrained for cell embedding task, and the cell type label fine-tuning is described in their publication. scMulan also uses the cell type label and cell condition label information in their pretraining task. We also agree that the incorporation of the cell labeling could lead to potential issue for underrepresented cell types. However, the contrastive loss serves as an regularization for the to correct for the batch effect that cannot be removed through masked token prediction (MTP). The higher resolution information will still be preserved by the model through the MTP task.

---

> ### Author Response · Authors · 2025-12-03
>
> ```
> Q7. Theorem 1 feels disconnected from the rest of the paper and the empirical results. How does Theorem 1 relate to the empirical findings of the paper and compressing genes using a pML?
> ```
> **A7**. We introduced Theorem 1 to provide theoretical analysis on why the design of gene compression model is efficient in compressing genes while retaining the key biological information. The proof shows that the min-cut loss upper-bound the biological information loss through compression. By minimizing the min-cut loss, the model is also reducing the information loss after the compression. We incorporate this proof to provide theorectical grounding of why the model achieve superior performance compared to baseline methods even after the compression.
>
> To connect the empirical findings to theorem 1, we conduct additional ablation tests by comparing the current compression design of scREBOUND with a self-learned compression model without the min-cut loss design. We construct two baseline scREBOUND models: 1. scREBOUND model with the compression model that is self-learned during training and removed of protein embedding information; 2. scREBOUND model with the compression model that is replaced by the simple highly-variable gene selection. We control the compressed dimension to be 256 for fair comparison. The comparison result (Tables 9-11) shows the efficiency our proposed gene compression model compared to the other designs.
>
> Table 9. Batch effect removal
> |Methods|Immune|||Pancreas|||Lung|||Covid19|||
> |-|-|-|-|-|-|-|-|-|-|-|-|-|
> ||ARI|NMI|ASW|ARI|NMI|ASW|ARI|NMI|ASW|ARI|NMI|ASW|
> |scREBOUND|0.57|0.75|0.61|0.60|0.74|0.60|0.52|0.70|0.58|0.46|0.63|0.64|
> |self-learned|0.25|0.40|0.48|0.18|0.33|0.44|0.13|0.26|0.35|0.06|0.17|0.46|
> |HVGs|0.10|0.22|0.47|0.23|0.41|0.48|0.05|0.20|0.42|0.04|0.15|0.47|
>
> Table 10. Cell type annotation with KNN
> |Method|Immune||Pancreas||Lung||Covid19||
> |-|-|-|-|-|-|-|-|-|
> ||WF1|Accu|WF1|Accu|WF1|Accu|WF1|Accu
> |scREBOUND|0.74|0.72|0.80|0.81|0.52|0.59|0.71|0.73|
> |self-learned|0.28|0.31|0.37|0.32|0.08|0.09|0.30|0.34|
> |HVGs|0.26|0.27|0.48|0.45|0.13|0.15|0.25|0.28|
>
> Table 11. Data imputation with 10% mask
> |Methods|Immune||Pancreas||Lung||Covid19||
> |-|-|-|-|-|-|-|-|-|
> ||SCorr|PCorr|SCorr|PCorr|SCorr|PCorr|SCorr|PCorr
> |scREBOUND|0.37|0.90|0.43|0.42|0.35|0.56|0.37|0.79|
> |self-learned|0.36|0.87|0.41|0.40|0.35|0.56|0.36|0.73|
>
> ```
> Q8. How was the cell type overlap between the 1 million randomly selected cells and the cells in the test set for zero-shot cell type annotation? Shouldn't several cell types not be represented in the randomly selected training set?
> ```
> **A8**. This is a great question. Firstly, we want to clarify that there is no universal standard for the cell type annotations. The cell type annotation is still relatively subjective, affected by factors including the focus of the experiments and the researchers naming conventions. The cell ontology information provides a way to align cell type annotations across studies into a standard ontology tree with annotation of different granularities. However, the alignment between the annotation from the study and the standard annotation in the cell ontology tree still need to be performed manually.
>
> The 1 million training datasets are sampled from the CellXGene repository, where the cell type annotation is already standardized using annotations from cell ontology tree. However, the cell type annotation in the test datasets are obtained from completely difference sources, and the annotation is not standardized. The first step we do is to standardize the annotation in the test datasets into the annotations in the cell ontology tree through manual mapping. We conducted the mapping with the help of biology experts.
>
> We intentionally select a large pool of cells for the training (1 million) to cover as many cell types that show up in CellXGene as possible. The 1 million sampled cell includes 676 standardized cell types, and all the standardized annotations in the test dataset have the corresponding cells in the sampled data. We then further select the cells in the training data that are under the test annotations pool, and train the classifier (SVM and KNN) on the selected training data. In this way, the test annotation labels pool completely aligned with the training labels pool.
>
> The "zero-shot" here means that the cell embedding generated by scREBOUND and baseline methods and used as the input of the classifier training is not further fine-tuned on the test dataset, but the classifier still needs to be trained for each test dataset separately.
>
> ```
> Q9. scGPTs' performance seems to improve with a higher percentage of masked genes for imputation (Table 6). Is there a potential reason for this?
> ```
> **A9**. The pattern in the original Table 6 was due to variance from a single run of scGPT. The issue disappear after updating the results to include multiple runs with averaged performance (see revised Table 4).

---

### Official Review · Reviewer_oADU · 2025-11-01

**Soundness:** 3
**Presentation:** 3
**Contribution:** 3
**Rating:** 6
**Confidence:** 4

**Summary:**

This paper presents scREBOUND, a computationally efficient foundation model for single-cell RNA-seq data that directly tackles two key challenges in current single-cell foundation models (scFMs): (1) high computational cost, and (2) poor handling of batch effects. The authors propose (a) a biologically informed gene compression module (protein language model–guided mincut-based grouping of genes into 256 meta-genes), (b) a batch encoder capturing batch-specific features, and (c) a multi-granular contrastive loss leveraging hierarchical cell ontologies to align cell types across batches of varying label granularity. scREBOUND is pretrained with masked token prediction and evaluated on multiple downstream zero-shot tasks: batch effect removal, cell type annotation, cross-species knowledge transfer, and missing value imputation. Results show that scREBOUND achieves competitive or superior accuracy compared to state-of-the-art single-cell foundation models (scGPT, UCE, scMulan, scFoundation), while reducing inference runtime and memory usage.

**Strengths:**

- The paper successfully tackles the computational bottleneck of scFMs. scREBOUND achieves significant advantages in runtime and GPU memory consumption.

-  The use of a protein-language-model-informed gene compression strategy is novel. By reducing the gene tokens to 256 meta-genes, the model reduces the number of tokens processed by the transformer while retaining key biological features. Furthermore, the paper provides a theoretical analysis showing that the mincut regularized compression strategy upper-bounds the mutual information loss after feature compression.

-  The combined approach of the batch embedding network and the multi-granular contrastive loss effectively addresses the systematic noise inherent in scRNA-seq data.

- scREBOUND consistently performs well across a diverse suite of zero-shot tasks. It demonstrated top performance in batch effect removal across all eight test datasets.

**Weaknesses:**

-All the main results are framed in a purely zero-shot setting. In practice, users often do a light finetuning or task-adaptive pretraining step, and several influential models in this space (such as Geneformer) are typically reported with some form of adaptation. Because the paper does not compare against Geneformer or against scGPT/scFoundation in a finetuned regime, it’s unclear whether scREBOUND would still outperform strong baselines once they are allowed to adapt to the target dataset.

- A second, more structural point: the gene–meta-gene compression is built from protein-language-model embeddings of genes. That’s elegant, but protein-embedding similarity does not necessarily reflect transcriptional or regulatory co-function — e.g. co-expression, pathway co-membership, TF–target relationships, ligand–receptor pairs, or chromatin-linked interactions may never be close in protein-embedding space. So the current graph might be biased toward sequence/structure similarity rather than cellular co-usage. This clustering stage could optionally ingest an external gene–gene interaction network (STRING, BioGRID, Reactome functional edges, or even dataset-specific co-expression graphs).

**Questions:**

- The authors adopt Fourier encoding to transform continuous expression values into embeddings, citing poor MLP performance on long-tailed gene expression distributions. Could the authors briefly elaborate on why Fourier encoding was ultimately preferred, given that scREBOUND’s design emphasizes computational efficiency?

-  The compression module reduces all input genes to 256 meta-genes. Was this specific number chosen through empirical tuning (e.g., grid search on performance vs. efficiency), or is there a theoretical motivation related to the information content or redundancy in scRNA-seq data?

-  The meta-gene construction relies on a gene graph induced from protein LM embeddings. But protein similarity does not necessarily capture transcriptional co-regulation, TF–target pairs, or pathway-level co-usage. Could your method accept an external gene–gene interaction network (e.g. STRING/BioGRID/co-expression from the same atlas) as an additional adjacency or regularizer in the mincut step?

---

> ### Author Response · Authors · 2025-12-02
>
> ```
> Q1. All the main results are framed in a purely zero-shot setting. In practice, users often do a light finetuning or task-adaptive pretraining step, and several influential models in this space (such as Geneformer) are typically reported with some form of adaptation. Because the paper does not compare against Geneformer or against scGPT/scFoundation in a finetuned regime, it’s unclear whether scREBOUND would still outperform strong baselines once they are allowed to adapt to the target dataset.
> ```
> **A1**. We agree with the reviewer that light fine-tuning is widely used in many practical applications, in addition to directly applying foundation models in a zero-shot manner. Following the reviewer’s suggestion, we have incorporated fine-tuned versions of scREBOUND, scGPT, and scFoundation into our evaluation. We did not include fine-tuning for the other baseline models because detailed fine-tuning procedures are not provided for those methods.
>
> We evaluated the fine-tuned models on three major benchmarking tasks: (1) batch effect removal, (2) cell type annotation, and (3) data imputation. The corresponding results are presented in Tables 1~3. scREBOUND still shows superior performance in the fine-tuned setting.
>
> Table 1. The evaluation of batch effect removal on fine-tuned models
> |Methods|Immune|||||Pancreas|||||Lung|||||Covid19|||||CL|||||HumanPBMC|||||PancRM|||||PBMC_Raw|||||
> |-|-|-|-|-|-|-|-|-|-|-|-|-|-|-|-|-|-|-|-|-|-|-|-|-|-|-|-|-|-|-|-|-|-|-|-|-|-|-|-|-|
> ||ARI|NMI|ASW|ASW-Batch|GC|ARI|NMI|ASW|ASW-Batch|GC|ARI|NMI|ASW|ASW-Batch|GC|ARI|NMI|ASW|ASW-Batch|GC|ARI|NMI|ASW|ASW-Batch|GC|ARI|NMI|ASW|ASW-Batch|GC|ARI|NMI|ASW|ASW-Batch|GC|ARI|NMI|ASW|ASW-Batch|GC|
> |scREBOUND-finetune|0.61|0.76|0.61|0.77|0.70|0.66|0.69|0.61|0.81|0.82|0.56|0.70|0.58|0.79|0.67|0.63|0.69|0.66|0.70|0.49|0.69|0.71|0.88|0.74|1.00|0.41|0.64|0.64|0.69|0.86|0.43|0.57|0.54|0.74|0.88|0.74|0.72|0.60|0.84|0.88|
> |scGPT-finetune|0.54|0.68|0.53|0.83|0.88|0.26|0.44|0.47|0.82|0.89|0.51|0.66|0.56|0.80|0.75|0.63|0.63|0.58|0.73|0.63|0.41|0.43|0.58|0.87|1.00|0.32|0.41|0.46|0.85|1.00|0.20|0.45|0.35|0.57|0.88|0.26|0.34|0.51|0.82|1.00|
> |scFoundation-finetune|0.56|0.74|0.61|0.79|0.81|0.53|0.65|0.56|0.84|0.93|0.49|0.71|0.57|0.80|0.79|0.37|0.60|0.59|0.76|0.69|0.76|0.66|0.58|0.91|1.00|0.29|0.42|0.47|0.79|1.00|0.20|0.46|0.38|0.56|0.92|0.10|0.22|0.49|0.89|0.99|
>
> Table 2. The evaluation of cell type annotation on fine-tuned models
> |Methods|Immune|||Pancreas|||Lung|||Covid19|||HumanPBMC|||PancRM|||PBMC|||
> |-|-|-|-|-|-|-|-|-|-|-|-|-|-|-|-|-|-|-|-|-|-|
> ||Acc|F1(Unw)|F1(W)|Acc|F1(Unw)|F1(W)|Acc|F1(Unw)|F1(W)|Acc|F1(Unw)|F1(W)|Acc|F1(Unw)|F1(W)|Acc|F1(Unw)|F1(W)|Acc|F1(Unw)|F1(W)|
> |scREBOUND-finetune|0.81|0.77|0.79|0.84|0.63|0.80|0.78|0.72|0.78|0.91|0.91|0.91|0.92|0.82|0.92|0.81|0.61|0.75|0.92|0.85|0.92|
> |scFoundation-finetune|0.88|0.82|0.85|0.87|0.72|0.82|0.92|0.89|0.92|0.95|0.95|0.95|0.98|0.95|0.98|0.91|0.80|0.89|0.94|0.90|0.94|
> |scGPT-finetune|0.78|0.64|0.75|0.49|0.23|0.43|0.68|0.67|0.63|0.92|0.92|0.93|0.47|0.33|0.37|0.19 |0.04|0.11|0.26|0.12|0.21|
> |scMulan-finetune|0.86|0.80|0.83|0.85|0.71|0.81|0.91|0.88|0.92|0.95|0.95|0.95|0.97|0.93|0.97|0.89|0.76|0.86|0.92|0.85|0.92|
>
> Table 3. The evaluation of data imputation on fine-tuned models
> |Mask|Method|Immune||Pancreas||Lung||Covid19||CL||HumanPBMC||PancRM||PBMC_Raw||
> |-|-|-|-|-|-|-|-|-|-|-|-|-|-|-|-|-|-|
> |||Pcorr|Scorr|Pcorr|Scorr|Pcorr|Scorr|Pcorr|Scorr|Pcorr|Scorr|Pcorr|Scorr|Pcorr|Scorr|Pcorr|Scorr|
> |10%|scREBOUND-finetune|0.9314|0.3841|0.6250|0.4986|0.7298|0.3950|0.8223|0.3734|0.9661|0.5585|0.9414|0.3863|0.6615|0.5302|0.8852|0.2736|
> ||scGPT-finetune|-0.0002|-0.0002|0.0001|0.0002|-0.0002|-0.0007|0.0003|-0.0003|0.0010|0.0008|-0.0001|0.0004|0.0005|0.0008|-0.0004|-0.0011|
> |20%|scREBOUND-finetune|0.9300|0.3860|0.6274|0.4979|0.7484|0.4002|0.8367|0.3739|0.9679|0.5603|0.9441|0.3897|0.6727|0.5268|0.8943|0.2739|
> ||scGPT-finetune|-0.0003|-0.0003|0.0004|-0.0001|-0.0002|-0.0007|0.0000|-0.0004|0.0005|0.0004|0.0002|0.0004|0.0005|0.0008|-0.0004|-0.0010|

---

> ### Author Response · Authors · 2025-12-02
>
> ```
> Q2.  second, more structural point: the gene–meta-gene compression is built from protein-language-model embeddings of genes. That’s elegant, but protein-embedding similarity does not necessarily reflect transcriptional or regulatory co-function — e.g. co-expression, pathway co-membership, TF–target relationships, ligand–receptor pairs, or chromatin-linked interactions may never be close in protein-embedding space. So the current graph might be biased toward sequence/structure similarity rather than cellular co-usage. This clustering stage could optionally ingest an external gene–gene interaction network (STRING, BioGRID, Reactome functional edges, or even dataset-specific co-expression graphs).
> ```
> **A2**. We appreciate the reviewer’s insightful comment. Indeed, protein–embedding–based similarity captures aspects of sequence and structural relatedness, which may not always align with transcriptional co-regulation or pathway-level co-function. While this raises a reasonable concern, prior studies have shown that protein language model embeddings are among the strongest general-purpose representations of gene function across diverse tasks [1, 2].
>
> To directly evaluate whether protein embeddings introduce a systematic bias in our compression stage, we conducted an additional experiment in which we replaced the protein–embedding–based graph with a graph constructed from Gene Ontology (GO) functional similarities. Even though GO are not broadly generalizable across all genes in all species compared to protein language model, it still is the golden standard for most gene's functional annotation.
>
> The compression architecture follows the protein embedding compression architecture, but the token embedding is replaced with GO annotation vector and the gene-gene similarity graph is constructed using GO term similarities. For a fair comparison, we fix the compressed number of tokens to be 256, same as the compression unit that we used. We select four test datasets, including Immune, Pancrease, Lung, and Covid19. We pretrain the model and compare the model performance on tasks including batch effect removal, cell type annotation, and imputation.
>
> The result (Tables 4-6) shows that compression with protein-embedding graph performs comparably with GO graph, and does not introduce a meaningful functional bias into the compressed gene representation.
>
> **Table 4**: Performance of different compression framework on batch effect removal
> |Methods|Immune|||Pancreas|||Lung|||Covid19|||
> |-|-|-|-|-|-|-|-|-|-|-|-|-|
> ||ARI|NMI|ASW|ARI|NMI|ASW|ARI|NMI|ASW|ARI|NMI|ASW|
> |scREBOUND|0.57|0.75|0.61|0.60|0.74|0.60|0.52|0.70|0.58|0.46|0.63|0.64|
> |GO|0.67|0.75|0.61|0.62|0.76|0.62|0.52|0.71|0.59|0.44|0.61|0.64|
>
> **Table 5**: Performance of different compression framework on cell type annotation
> |Classifier|Methods|Immune||Pancreas||Lung||Covid19||
> |-|-|-|-|-|-|-|-|-|-|
> |||WF1|Accu|WF1|Accu|WF1|Accu|WF1|Accu|
> |kNN|scREBOUND|0.74|0.72|0.80|0.81|0.52|0.59|0.71|0.73|
> ||GO|0.75|0.78|0.83|0.84|0.52|0.61|0.72|0.73|
> |SVM|scREBOUND|0.78|0.79|0.90|0.89|0.54|0.58|0.79|0.79|
> ||GO|0.73|0.78|0.87|0.88|0.56|0.58|0.82|0.82|
>
> **Table 6**: Performance of different compression framework on data imputation
> |Mask%|Methods|Immune||Pancreas||Lung||Covid19||
> |-|-|-|-|-|-|-|-|-|-|
> |||SCorr|PCorr|SCorr|PCorr|SCorr|PCorr|SCorr|PCorr|
> |10%|scREBOUND|0.37|0.90|0.43|0.42|0.35|0.56|0.37|0.79|
> |10%|GO|0.37|0.90|0.43|0.42|0.37|0.60|0.37|0.79|
> |20%|scREBOUND|0.35|0.84|0.40|0.34|0.35|0.56|0.35|0.74|
> |20%|GO|0.35|0.84|0.40|0.38|0.35|0.55|0.35|0.74|
>
> 1. Rosen, Yanay, et al. "Toward universal cell embeddings: integrating single-cell RNA-seq datasets across species with SATURN." Nature Methods 21.8 (2024): 1492-1500.
> 2. Rosen, Yanay, et al. "Universal cell embeddings: A foundation model for cell biology." bioRxiv (2023): 2023-11.
> ```
> Q3. The authors adopt Fourier encoding to transform continuous expression values into embeddings, citing poor MLP performance on long-tailed gene expression distributions. Could the authors briefly elaborate on why Fourier encoding was ultimately preferred, given that scREBOUND’s design emphasizes computational efficiency?
> ```
> **A3**. As we have stated in the manuscript, there are mainly three ways for encoding the continuous gene expression value in single-cell foundation models: (1) Digitizing expression value $e_{i,m_j}$ and learns embedding for each digit; (2) Feeding $e_{i,m_j}$ into Multilayer Perceptron (MLP); and (3) Feeding $e_{i,m_j}$ into Fourier encoding function. Method (1) prevents the gradient from back-propagating to the gene compression model, and method (2) does not work well with the long-tail gene expression level distribution. Making the first two ways undesirable for the design of scREBOUND. Fourier transform, on the other hand, solve both issue. In addition, the computational time consumption of fourier is not significant when there are only 256 gene. That is why we adopted fourier transform in the end.

---

> ### Author Response · Authors · 2025-12-02
>
> ```
> Q4. The compression module reduces all input genes to 256 meta-genes. Was this specific number chosen through empirical tuning (e.g., grid search on performance vs. efficiency), or is there a theoretical motivation related to the information content or redundancy in scRNA-seq data?
> ```
> **A4**. 256 is a number that we picked considering the trade-off between the model performance and the running speed. More meta-genes will likely lead to higher precision but increase running time, but fewer meta-genes may cause too much information loss.
>
> We conduct comprehensive hyper-parameter test on scREBOUND with 64, 128, 256, 300, and 400 meta-genes, and select 4 test datasets for the evaluation. The results are shown in Tables 7-9. The training gets extremely slow and consumes significantly larger GPU memory with more than 400 meta-genes, and the advantage of scREBOUND, i.e. computational efficiency, would not be significant with too many meta-genes.
>
> Model with 300 and 400 meta-genes does not show significant performance gain compared to model with 256 meta-genes, but the inference time is significantly longer than the model with 256 meta-genes (e.g. model with 300 meta-genes has 1.4X of the inference time compared to 256 meta-genes). Model with 128 meta-genes and 64 meta-genes are faster than model with 256 meta-genes, but the performance is also the lower compared to the others especially in the tasks of cell type annotation and data imputation (Tables 2-3). The result shows that the number of meta-genes is selected based on the trade-off between inference efficiency and model performance. More meta-gene leads to better model performance, but slower inference speed.
>
> **Table 7**. Batch effect removal scores with different number of meta-genes
> |# tokens|Immune|||Pancreas|||Lung|||Covid19|||
> |-|-|-|-|-|-|-|-|-|-|-|-|-|
> ||ARI|NMI|ASW|ARI|NMI|ASW|ARI|NMI|ASW|ARI|NMI|ASW|
> |400|0.62|0.77|0.62|0.48|0.67|0.56|0.49|0.71|0.58|0.45|0.63|0.65|
> |300|0.62|0.76|0.62|0.49|0.69|0.58|0.47|0.70|0.57|0.46|0.62|0.64|
> |256|0.57|0.75|0.61|0.60|0.74|0.60|0.52|0.70|0.58|0.46|0.63|0.64|
> |128|0.75|0.79|0.60|0.49|0.69|0.59|0.47|0.69|0.57|0.45|0.62|0.64|
> |64|0.77|0.78|0.62|0.63|0.73|0.59|0.47|0.69|0.55|0.45|0.61|0.63|
>
> **Table 8**. Cell type annotation scores with different number of meta-genes
> |Classifier|# tokens|Immune||Pancreas||Lung||Covid19||
> |-|-|-|-|-|-|-|-|-|-|
> |||WF1|Accu|WF1|Accu|WF1|Accu|WF1|Accu|
> |kNN|400|0.65|0.63|0.68|0.69|0.51|0.58|0.71|0.73|
> ||300|0.72|0.70|0.75|0.76|0.51|0.59|0.69|0.70|
> ||256|0.74|0.72|0.80|0.81|0.52|0.59|0.71|0.73|
> ||128|0.74|0.73|0.74|0.76|0.51|0.58|0.69|0.70|
> ||64|0.71|0.68|0.79|0.80|0.45|0.53|0.71|0.72|
> |SVM|400|0.75|0.73|0.78|0.77|0.49|0.52|0.81|0.82|
> ||300|0.78|0.77|0.83|0.82|0.51|0.54|0.80|0.81|
> ||256|0.78|0.79|0.90|0.89|0.54|0.58|0.79|0.79|
> ||128|0.74|0.73|0.81|0.80|0.51|0.55|0.79|0.80|
> ||64|0.76|0.64|0.81|0.80|0.43|0.49|0.79|0.79|
>
> **Table 9**. Data imputation with different number of meta-genes
> |Mask %|# tokens|Immune||Pancreas||Lung||Covid19||
> |-|-|-|-|-|-|-|-|-|-|
> |||SCorr|PCorr|SCorr|PCorr|SCorr|PCorr|SCorr|PCorr|
> |10%|400|0.36|0.88|0.42|0.40|0.37|0.59|0.36|0.77|
> |10%|300|0.37|0.89|0.42|0.43|0.37|0.60|0.36|0.78|
> |10%|256|0.37|0.90|0.43|0.42|0.35|0.56|0.37|0.79|
> |10%|128|0.36|0.87|0.41|0.36|0.37|0.57|0.36|0.75|
> |10%|64|0.37|0.90|0.43|0.44|0.36|0.56|0.36|0.78|
> |20%|400|0.34|0.83|0.39|0.32|0.34|0.54|0.34|0.73|
> |20%|300|0.34|0.84|0.39|0.36|0.35|0.55|0.34|0.74|
> |20%|256|0.35|0.84|0.40|0.34|0.35|0.56|0.35|0.74|
> |20%|128|0.34|0.81|0.38|0.29|0.34|0.53|0.34|0.71|
> |20%|64|0.35|0.84|0.40|0.38|0.34|0.51|0.34|0.74|
>
>
> ```
> Q5. The meta-gene construction relies on a gene graph induced from protein LM embeddings. But protein similarity does not necessarily capture transcriptional co-regulation, TF–target pairs, or pathway-level co-usage. Could your method accept an external gene–gene interaction network (e.g. STRING/BioGRID/co-expression from the same atlas) as an additional adjacency or regularizer in the mincut step?
> ```
> **A5**. Thank you for the thoughtful question. As discussed in A2, we use protein language model embeddings because they are the most broadly applicable gene representations across genes and species. Our experiment in A2 shows that protein-based compression performs comparable to GO-based compression (GO is the gold standard for functional annotation), suggesting that protein embeddings do not introduce meaningful bias.
>
> At the same time, scREBOUND is fully compatible with alternative gene–gene graphs. In A2, we demonstrated this by directly replacing the protein-embedding graph with a GO-informed graph without changing the architecture. Likewise, external resources such as STRING, BioGRID, Reactome, or dataset-specific co-expression graphs can be used as the adjacency or regularizer in the mincut step if desired.

---

### Meta-Review · Area_Chair_x1cm · 2026-01-10

**Summary:**

scREBOUND is a single-cell foundation model that compresses gene tokens into meta-genes and adds batch-aware training (batch encoder and multi-granular supervised contrastive loss) to improve robustness to batch effects. Reviews converges on two decision-critical concerns.

One reviewer argues that the core “pLM-guided meta-gene compression” idea is not new and was introduced earlier (and not cited), which weakens the contribution. Relatedly, several reviewers felt the paper’s strongest gains appear to come from the supervised contrastive objective (which uses labels/ontology during pretraining), raising questions about how much is due to the proposed compression/batch design versus adding supervision that some baselines do not use in the same way.

The authors added several missing pieces (fine-tuning comparisons, batch integration metrics, error bars, SATURN comparison, feature ablations), which helps. But reviewers still raised concerns about (i) missing key baselines (notably Geneformer in any setting), (ii) comparability of fine-tuning across methods (procedures not uniformly available), and (iii) some results that look implausible/bug-prone (e.g., near-zero or negative correlations for scGPT imputation across datasets), which undermines confidence in the empirical claims.

**Reviewer Concerns:**

Novelty and missing citations (UZvz): The rebuttal does not convincingly resolve the concern that pLM-informed meta-gene compression was previously proposed and should be properly cited and differentiated. The SATURN comparison helps, but it does not address the broader novelty claim.

Fairness of supervision in pretraining (UZvz, partially GCU6): The paper’s multi-granular contrastive loss uses ontology/labels during pretraining. Even if some baselines use labels in some stage, the supervision structure here appears central and may not be matched by the compared baselines. The rebuttal clarifies intent, but it does not fully establish that comparisons are “apples-to-apples,” nor does it quantify performance without this supervision in a way that cleanly supports the main claims.

Key baseline coverage (oADU): Geneformer is still not included. Since Geneformer is widely used with task-adaptive pretraining or fine-tuning, its absence leaves a meaningful gap in the practical comparison. Some baselines behave strangely (especially imputation numbers for scGPT that are ~0 or negative across datasets). The rebuttal says one pattern was due to variance, but the overall magnitude/behavior still looks suspect and is not convincingly audited (e.g., consistent preprocessing, masking protocol, or evaluation pipeline checks).

**Reviewer Scores:**

Reviewer oADU: Many of their concrete questions were addressed (fine-tuning, GO graph alternative). However, discussion with other reviewers would likely elevate the novelty/fairness concerns and the missing Geneformer baseline. That pressure typically nudges a marginal accept toward borderline.

Reviewer UZvz: The rebuttal substantively addressed several of their biggest technical complaints (added batch integration metrics, error bars, SATURN comparison, fine-tuning, batch-feature ablation, theorem-to-empirics connection). Even so, their core objection about limited novelty and evaluation trustworthiness/fairness would likely remain, and that is usually decisive for a strong reject.

Reviewer GCU6: Their main questions were answered.

---

### Decision · Program_Chairs · 2026-01-26

Reject